



# Is the Blade Element Momentum Theory overestimating Wind Turbine Loads? – A Comparison with a Lifting Line Free Vortex Wake Method

Perez-Becker Sebastian[1], Papi Francesco[2], Saverin Joseph[1], Marten David[1], Bianchini Alessandro[2], and Paschereit Christian Oliver[1]

[1]Chair of Fluid Dynamics, Hermann Föttinger Institute, Technische Universität Berlin, Berlin, Germany
[2]Department of Industrial Engineering, Università degli Studi di Firenze, Florence, Italy

**Correspondence:** Sebastian Perez-Becker (s.perez-becker@fd.tu-berlin.de)

**Abstract.** Load calculations play a key role in determining the design loads of different wind turbine components. State of the art in the industry is to use the Blade Element Momentum (BEM) theory to calculate the aerodynamic loads. Due to their simplifying assumptions of the rotor aerodynamics, BEM methods have to rely on several engineering correction models to capture the aerodynamic phenomena present in Design Load Cases (DLCs) with turbulent wind. Because of this, BEM methods

can overestimate aerodynamic loads under challenging conditions when compared to higher-order aerodynamic methods – such as the Lifting Line Free Vortex Wake (LLFVW) method – leading to unnecessarily high design loads and component costs. In this paper, we give a quantitative answer to the question of BEM load overestimation by comparing the results of aeroelastic load calculations done with the BEM-based OpenFAST code and the QBlade code which uses a LLFVW method. We compare extreme and fatigue load predictions from both codes using 66 ten-minute load simulations of the DTU 10 MW Reference

Wind Turbine according to the IEC 61400-1 power production DLC group.

Results from both codes show differences in fatigue and extreme load estimations for practically all considered sensors of the turbine. LLFVW simulations predict 4% and 14% lower lifetime Damage Equivalent Loads (DELs) for the out-of-plane blade root and the tower base fore-aft bending moments, when compared to BEM simulations. The results also show that lifetime DELs for the yaw bearing tilt- and yaw moments are 2% and 4% higher when calculated with the LLFVW code. An

ultimate state analysis shows that extreme loads of the blade root out-of-plane and the tower base fore-aft bending moments predicted by the LLFVW simulations are 3% and 8% lower than the moments predicted by BEM simulations, respectively. Further analysis reveals that there are two main contributors to these load differences. The first is the different treatment in both codes of the effect that sheared inflow has on the local blade aerodynamics and second is the wake memory effect model which was not included in the BEM simulations.

## 1   Introduction

Load calculations are an essential process when designing large modern wind turbines. With the help of such simulations, turbine designers are able to derive the design loads for each of the turbine's components. International guidelines and standards





prescribe for each load calculation loop a large number of aeroelastic simulations of the complete turbine (IEC 61400-1 Ed. 3). These simulations, or Design Load Cases (DLCs), are required in order to cover many possible situations that the wind turbine might encounter in its lifetime and hence calculate realistic loads. In the case of turbulent wind simulations, several repetitions of individual DLCs for a given configuration are required to limit the effect of statistical outliers and obtain converged results.

The current industry trend is to design ever larger wind turbines with increasingly long and slender blades. As the wind turbines become larger, the design loads of each component scale accordingly (Jamieson, 2018). This leads to increased material requirements and ultimately to higher component costs. Given this fact, there is a large incentive to calculate the components' loads as accurately as possible. Differences in load estimations on these large, multi MW scales can result in a considerable reduction in material use and consequently component costs.

Current aeroelastic codes rely mostly on the Blade Element Momentum (BEM) aerodynamic model (Hansen, 2008; Burton et al., 2011) to calculate aerodynamic loads. BEM models are computationally inexpensive but require a series of engineering corrections to model the more challenging unsteady aerodynamic phenomena usually present in the DLCs. This can lead to inaccurate predictions of the turbine's design loads. The advantages of BEM methods have become less compelling because of the increase in available computational power. For the same reason, methods with higher-order representations of the unsteady

aerodynamics have become more attractive. Vortex methods such as the Lifting Line Free Vortex Wake (LLFVW) aerodynamic model are able to model the turbine wake and its interaction with the turbine directly instead of relying on momentum balance equations – as BEM models do. Therefore, LLFVW models are able to calculate unsteady aerodynamics with far less assumptions than BEM models (Hauptmann et al., 2014; Perez-Becker et al., 2018). Using more accurate aerodynamic methods lowers model uncertainty, potentially lowering design loads and safety factors and ultimately leading to more competitive

turbine designs.

    Over the past years, there have been several studies comparing BEM models with higher-order vortex models. Gupta and Leishman compare the performance coefficients of a small two-bladed wind turbine using a BEM and a LLFVW method (Gupta and Leishman, 2005). They find that for scenarios in which the Tip Speed Ratio (TSR) is above the optimal TSR or in which there is a high yawed inflow, the BEM and LLFVW methods predict different values of the thrust and power coefficients.

In (Madsen et al., 2012), Madsen et al. compare the predictions of several BEM-based codes, vortex-based codes and CFD-based codes. The authors simulate the NREL 5 MW Reference Wind Turbine (RWT) (Jonkman et al., 2009) for uniform and sheared wind inflow conditions at a wind speed of 8 m/s. They find that under uniform conditions, the considered codes predict similar power and thrust. This changes when sheared inflow conditions are simulated. Here, the differences in the predicted power, thrust and load variation between the codes are larger. In (Qiu et al., 2014), the authors present a LLFVW method and

analyze the unsteady aerodynamic loads in yawing and pitching procedures. They show that the load predictions from their method are closer to measured experimental data when compared to BEM calculations. In (Marten et al., 2015), Marten et al. use the LLFVW method implemented in the aeroelastic code QBlade (Marten et al., 2013b, a) to simulate the MEXICO (Snel et al., 2009) and the NREL Phase IV (Simms et al., 2001) experiments. They compare the results to experimental data and to predictions from other BEM and vortex codes, showing good agreement with the experimental results.





Several authors have also done aeroelastic comparative studies. In (Voutsinas et al., 2011), Voustinas et al. analyze the aeroelastic effect of sweeping a turbine blade backwards. For the NREL/UPWIND 5 MW RWT, they compare the loads predicted with a BEM method and GENUVP – a lifting surface method coupled with a vortex particle representation of the wake (Voutsinas, 2006). Their study concludes that the BEM method underestimates the power reduction arising from the bend-

twist coupling. Jeong et al. extended the study from (Madsen et al., 2012) by considering flexibility in their turbine model as well as inflow conditions with turbulent wind (Jeong et al., 2014). They find that for lower wind speeds, there are noticeable differences in the predicted loads from BEM and LLFVW methods. For higher wind speeds though, these differences decrease due to the overall smaller axial induction factors. In (Gebhardt and Roccia, 2014), the authors present an aeroelastic tool for wind energy applications. It has a flexible structural model that can combine rigid-body dynamics, assumed-mode techniques

and finite element methods. Their model uses a lifting surface method combined with free vortex wake method to calculate the aerodynamic loads. The authors compare the power prediction of a three-bladed wind turbine using their method and a BEM method when the turbine sees yawed inflow, showing considerable differences in their predictions.

Other comparisons of vortex and BEM methods are done in (Hauptmann et al., 2014; Boorsma et al., 2016). Here, the authors compare the aeroelastic predictions of LLFVW and BEM methods for several load cases. Both studies share the same

LLFVW method named AWSM (Van Garrel, 2003) but use different structural codes. The studied cases include a pitch fault scenario, extreme coherent gust with direction change simulations, yawed inflow, turbine in half wake, wind shear and turbulent wind conditions. Both studies conclude that for their considered cases the LLFVW predicts lower load fluctuations. Chen et al. perform a study of the NREL 5 MW RWT considering yawed and shared inflow using a free wake lifting surface model and a geometrically exact beam model (Chen et al., 2018). They find that the yawed inflow model used in the BEM simulations

overpredicts the variation in the induced velocity when compared to their vortex method. Saverin et al. couple in (Saverin et al., 2016a) the LLFVW method from QBlade to the structural code of FAST (Jonkman and Buhl, 2005). The authors use the NREL 5 MW RWT and compare the loads predicted by the LLFVW method and AeroDyn – the BEM code used in FAST (Moriarty and Hansen, 2005) – showing significant differences in loading and controller behavior. Large differences can also be seen in (Saverin et al., 2016b). Here, Saverin et al. combine QBlade's LLFVW method and a structural model with a geometrically

exact beam model for the rotor blade. The authors simulate the DTU 10 MW RWT (Bak et al., 2013) under steady uniform wind conditions and the emergency stop load case. Load case simulations are also performed by Perez-Becker et al. in (Perez-Becker et al., 2018). Here, the authors simulate the DTU 10 MW RWT in power production DLCs as defined in (IEC 61400-1 Ed. 3) including wind shear, yaw error and turbulent inflow conditions. They conclude that for wind speeds above rated wind, the BEM-based aeroelastic code FAST predicts higher fatigue loading and pitch activity that the LLFVW-based code QBlade.

A higher fidelity code is presented in (Sessarego et al., 2017). In this work, Sessarego et al. couple the structural model of FLEX5 to the viscous-inviscid interactive aerodynamic method MIRAS to obtain an aeroelastic code. The authors compare this code to several BEM-based codes in load cases including steady uniform wind, uniform yawed inflow and turbulent inflow. A hybrid implementation of BEM method for the far wake and a LLFVW method for the near wake is presented in (Pirrung et al., 2017). Here, Pirrung et al. compare the predictions of their hybrid near-wake model to a pure BEM method and the

lifting-surface free-wake method GENUVP. Results from pitch step responses and prescribed vibration cases for the NREL 5





MW RWT show that the near-wake method agrees much better with the lifting-surface free-wake method that with the pure BEM method.

So far, most of the studies comparing loads have focused on specific scenarios, simulating turbines under idealized inflow conditions or using a small number of turbulent load cases. If we wish to answer quantitatively how the results of load cal-
culations differ when we use BEM-based and LLFVW-based methods, we need a large number of turbulent DLCs to level out statistical biases of individual realizations. Many of the mentioned studies also do not include the direct interaction with the turbine controller. Wind turbine load calculations are aero-servo-elastic in nature and the predicted loads are a result of the interaction of the aerodynamics with the turbine structure and controller. Not taking this interaction into account gives an incomplete picture of the effect that different aerodynamic models have on the design loads of the wind turbine.

In this paper, we compare the results of aero-servo-elastic load calculations for the DTU 10 MW RWT. The turbine is simulated according to the IEC 61400-1 ed.3 DLC groups 1.1 and 1.2 using two different aeroelastic codes: NREL's BEM-based OpenFAST (OpenFAST) and TU Berlin's LLFVW-based QBlade. Fatigue and extreme loads of key turbine sensors, derived from 66 ten-minute simulations covering a wind speed range between 4 m/s and 24 m/s, are compared and analyzed. Section 2 gives an overview of the aerodynamic and structural codes as well as the controller used in this study. A baseline
comparison of the codes under idealized inflow conditions is done in Sect. 3, where we compare the performance of our turbine when calculated with both codes. Sections 4 to 6 contain the main contribution of this paper: a comparison and analysis of the results of load calculations with turbulent wind using both codes. Section 4 presents the considered sensors and gives an overview of the results. Section 5 presents, analyses and discusses the fatigue loads. An ultimate load analysis including discussion is presented in Sect. 6 and the conclusions are drawn in Sect. 7.

## 2 Methods

For this study, we chose to use the DTU 10 MW RWT. It is representative of the new generation of wind turbines and has been used in several research studies. The complete description of the turbine can be found in (Bak et al., 2013).

The following subsections briefly present the methods used for aerodynamic and structural modeling, the turbine controller and the setup used for the load simulations.

### 2.1 Aerodynamic Models

OpenFAST and QBlade are set up so that their only difference is the implemented aerodynamic model. OpenFAST uses a BEM method and QBlade a LLFVW method.

#### 2.1.1 Blade Element Momentum-Method

The BEM method calculates the aerodynamic loads by combining the blade element theory and the momentum theory of an
actuator disc to obtain the induced velocities on every discretized element of the blades (Moriarty and Hansen, 2005). The turbine rotor is divided into independently-acting annuli. For each annulus, the thrust and torque obtained from 2D airfoil polar





data of the blade element is equated to the thrust and torque derived from the momentum theory of an actuator disc (Burton et al., 2011). This set of equations can be solved iteratively to obtain the forces and moments on each blade element. This theory is only valid for uniform aligned flows in equilibrium. Several correction models have been developed to extend the BEM method so that more challenging aerodynamic situations can be modelled. The first five correction models implemented

in OpenFAST are described in (Moriarty and Hansen, 2005) and only briefly mentioned here.

   – **Tip- and root-loss**: This correction accounts for the finite number of blades that make up the rotor. OpenFAST uses the model developed by Prandtl that includes a radial dependent correction factor to the induced velocity of the rotor.

   – **Turbulent wake state**: This correction is included because the original BEM method fails to predict the turbulent wake mixing behind heavily-loaded rotors. This is accounted for in OpenFAST by including an empirical correction model

originally developed by Glauert (and adapted by Buhl) that modifies the thrust coefficient of the turbine rotor for high axial induction values.

   – **Oblique inflow**: This model accounts for the skewed wake shape when the turbine is in yawed-inflow condition. The model used in OpenFAST is based on the method developed by Pitt and Peters and modifies the local axial induction factor as a function of the blade radius, the rotor skew angle and the rotor azimuth angle.

– **Dynamic stall**: BEM codes include this correction to account for the unsteady aerodynamics on the blade element level. OpenFAST uses the dynamic stall model implemented by Beddoes and Leishman. This model modifies the static airfoil polar data to capture the unsteady effects in both attached and separated flow.

   – **Tower shadow**: In order to model the influence of the tower on blade aerodynamics, OpenFAST uses a potential flow model to account for the deficit of the incoming velocity in the region in front of the tower.

– **Wake memory effect**: This correction is needed to model the additional time required by the flow to adapt when sudden changes in pitch angle, rotational speed or wind speed occur at the rotor plane. This additional time comes from the interaction of the flow with the rotor wake. OpenFAST recently introduced this feature via the optional Dynamic BEM Theory (DBEMT) module. It uses one of the models presented in (Snel and Schepers, 1995) that filters the induced velocities via two first-order differential equations.

– **Stall delay**: Blade Element Theory assumes no interaction between the blade elements. For rotating airfoils in the inner part a wind turbine blade there is a significant amount of radial flow. This phenomenon delays the effective angle of attack at which the airfoil stalls (when compared to the 2D airfoil polar data). OpenFAST does not have an explicit model for stall delay. Instead, the airfoil polar data has to be pre-processed using an appropriate model before it is implemented in the code. For this study we used the 3D-corrected airfoil polar data presented in (Bak et al., 2013). The corrected airfoil

data was obtained using the method described in (Bak et al., 2006).





### 2.1.2 Lifting Line Free Vortex Wake-Method

The LLFVW-method is based on inviscid potential flow theory and a vortex representation of the flow field (Van Garrel, 2003; Marten et al., 2015). The rotor blade is discretized into elements represented by bound ring vortices. These bound vortices are located the quarter chord position and their sum make up a lifting line. By using the Kutta-Joukowsky theorem and the airfoil polar data corresponding to the blade element we can calculate the circulation of the bound vortices:

$$\Gamma = \frac{L}{|\boldsymbol{V_{tot}}|\rho} = C_l(\alpha)\frac{1}{2}|\boldsymbol{V_{tot}}| \cdot c. \tag{1}$$

In this equation, $\Gamma$ is the circulation of the blade element, $L$ is the lift per unit length, $\rho$ the density, $\boldsymbol{V_{tot}}$ the total velocity, $C_l$ the lift coefficient, $\alpha$ the angle of attack and $c$ the local chord. The total velocity is the sum of the incoming velocity $\boldsymbol{V_\infty}$, the velocity due to the motion of the blade (rotation / deflection) $\boldsymbol{V_{mot}}$ and the induced velocity from the wake $\boldsymbol{V_\Gamma}$:

$$\boldsymbol{V_{tot}} = \boldsymbol{V_\infty} + \boldsymbol{V_{mot}} + \boldsymbol{V_\Gamma}. \tag{2}$$

The induced velocity from all the vortex elements in the wake can be calculated by applying the Biot-Savart Law at each blade element:

$$\boldsymbol{V_\Gamma}(\boldsymbol{x}_p) = -\frac{1}{4\pi}\int \Gamma \frac{(\boldsymbol{x}_p - \boldsymbol{x}) \times d\boldsymbol{l}}{|\boldsymbol{x}_p - \boldsymbol{x}|^3}. \tag{3}$$

Here, $\boldsymbol{x}_p$ is the control point where the Biot-Savart Law is evaluated (e.g. the blade element), $\boldsymbol{x}$ is the position of each of the wake vortices and $d\boldsymbol{l}$ their vectorized length.

Equations (1 – 3) can be solved iteratively to obtain the circulation, the induced velocity and the forces at each blade element. At each time step, the circulation is shed to the wake creating trailing and shed vortices. The former arise from the spanwise variation of the circulation and the latter from the temporal variation. By applying Eq. (3) to the wake vortices, the free convection of the wake can be modelled. Figure 1 shows a closeup of a wind turbine blade during a LLFVW simulation using the aero-servo-elastic code QBlade. It includes the concepts explained in this section.

While capturing the flow physics of a wind turbine rotor much more accurately, LLFVW methods still use some correction models to account for all the aerodynamic phenomena present in turbulent load calculations.

- **Dynamic stall**: Because of the potential flow assumption and the use of airfoil polar data, a model is needed to account for the flow separation phenomenon. QBlade's LLFVW method uses the ATEFlap unsteady aerodynamic model (Bergami and Gaunaa, 2012), modified so that it excludes contribution of the wake in the attached flow region (Wendler et al., 2016).

- **Tower shadow**: The effect of the tower on the blade aerodynamics also has to be taken into account explicitly in the LLFVW simulations via an engineering model. QBlade uses the same potential flow model that is also used in OpenFAST (Bak et al., 2001).



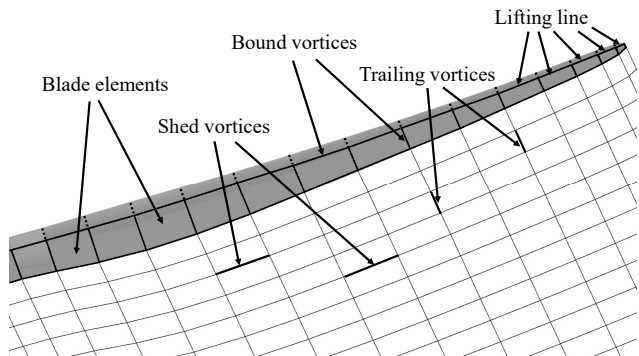

**Figure 1.** Representation of the LLFVW method and concepts on a wind turbine blade.

**Table 1.** Modeling differences of the two aerodynamic codes. I = Intrinsic; EM = Engineering Model

| Aerodynamic Phenomenon | BEM | LLFVW |
|---|---|---|
| Axial/Tangential induction | I | I |
| Radial induction | - | I |
| Tip- and root-loss | EM | I |
| Oblique inflow | EM | I |
| Turbulent wake state | EM | I |
| Wake memory effect | EM | I |
| Stall delay | EM | EM |
| Dynamic stall | EM | EM |
| Tower shadow | EM | EM |

- **Stall delay**: As with the BEM method, the stall delay phenomenon is included via modified airfoil polar data using an appropriate model. We used the same 3D-corrected airfoil polar data in both codes. The data was obtained with the method described in (Bak et al., 2006).

### 2.1.3 Comparison between the Aerodynamic Models

5   Table 1 summarizes the differences between the two aerodynamic models. The LLFVW method explicitly includes most of the phenomena present in DLC simulations with turbulent wind conditions. Usual DLC configurations include sheared and oblique inflow as well as temporal and spatial variations of the incoming wind speed. Unlike the BEM method that solves for the axial and tangential induction factors at each blade element, the LLFVW method solves for the complete flow around the rotor.





Turbine configurations include shaft tilt angles and blade cone angles. Including these angles, as well as the blade pre-bend and blade deflections in the case of aeroelastic calculations, violates the assumption made in BEM methods that the momentum balance takes place in the rotor plane. Thus, aerodynamic load predictions for the turbulent load cases obtained from LLFVW methods are expected to be more accurate compared to predictions from BEM methods. The radial induction mentioned in

Table 1 comes from the effect of the trailing vortices in the wake.

## 2.2   Structural Model

The structural model used for this study in both OpenFAST and QBlade is ElastoDyn (Jonkman, 2014). It uses a combined multi-body and modal dynamics representation that is able to model the wind turbine with flexible blades and tower (Jonkman, 2003). The modal representation of blades and tower uses an Euler-Bernoulli beam model to calculate deflections. The struc-

tural model allows for four tower modes: the first two fore-aft and side-side modes respectively. As for the blade, three modes are modelled in ElastoDyn: the first and second flapwise modes and the first edgewise mode. The structural model does not take into account shear deformation, axial-and torsional degrees of freedom.

Both OpenFAST and QBlade have additional models that allow for a more accurate representation of the wind turbine structural dynamics. The module BeamDyn in OpenFAST is able to model the blade as a geometrically exact beam (Wang

et al., 2016) and QBlade has a structural solver based on the open source multi-physics library CHRONO (Tasora et al., 2016). The latter uses a multi-body representation which includes Euler-Bernoulli beam elements in a co-rotational formulation. More accurate representations of the structural deflection of the wind turbine – in particular blade torsional deflection – have a significant influence on the loads. Torsional deflection changes the local angle of attack of a blade section and hence the lift force. This can lead to very different blade dynamics when compared to a model that does not include this degree of freedom.

Nonetheless, we decided to use ElastoDyn as the structural model for our study. It is shared by both aeroelastic codes so by using it, we keep the modeling differences only in the aerodynamic module and ensure that the latter is the only source of the load differences.

## 2.3   Controller

To enable aero-servo-elastic studies, we implemented a wind turbine controller that is compatible with both codes. The con-

troller is based on the DTU Wind Energy Controller (Hansen et al., 2013), which features pitch and torque control. It has been extended with a supervisory control based on a report by Iribas et al. (Iribas et al., 2015). The supervisory control enables the controller to run a full load analysis. The controller parameters were taken from the report (Borg et al., 2015). Only the optimal torque-speed gain was recalculated based on the maximum power coefficient obtained from OpenFAST calculations.

The controller parameters were obtained via BEM calculations, so it is expected that the controller will behave differently if

used in LLFVW calculations. We deliberately did not re-tune the controller parameters for the LLFVW simulations. This way, load differences arise not only from the different aerodynamic models themselves but also from the interaction of identical turbine controllers with these aerodynamic models. This procedure mimics to some extent current industry standards, in which




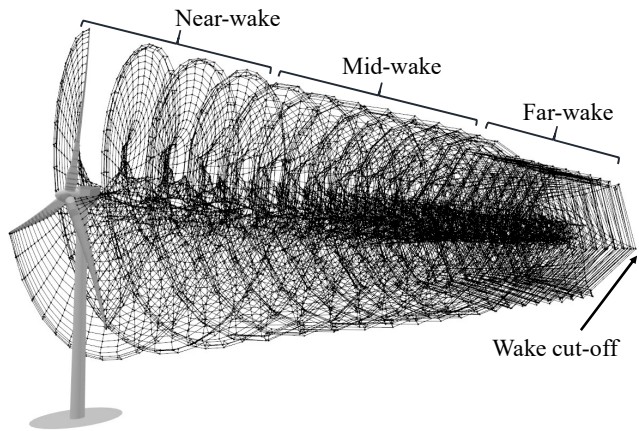

**Figure 2.** Wake coarsening methods for the LLFVW simulations: The wake is split into three regions with decreasing amount of wake elements. After a given number of revolutions, the wake is cut off.

wind turbine controllers are often pre-tuned using BEM-based aero-servo-elastic codes before they are implemented in real wind turbines.

### 2.4 Practical Considerations for Load Calculations

In order to use the presented methods in load calculations, several practical considerations had to be taken into account.
Given that Eq. (3) has to be evaluated for each vortex element in the wake, calculating the convection of the wake can be computationally costly, slowing down the LLFVW calculations. In order to increase the calculation speed of these simulations, we implemented two wake coarsening methods. The first one follows a similar method as the one described in (Boorsma et al., 2018). Instead of skipping or removing vortices, the method implemented in QBlade lumps the wake elements together after a given number of rotor revolutions. The method reduces the number of vortex elements in the wake while conserving the total
vorticity. This is done in two stages, giving us three wake regions: the near-wake, the mid-wake and the far-wake. The number of vortices lumped together is given by a lumping factor. So QBlade uses two lumping factors: the mid-wake factor for the transition from near-wake to mid-wake and the far-wake factor for the transition from mid-wake to far-wake.

The second method is the wake cut-off. After a given amount of rotor revolutions, the wake is cut off. The influence of these far-wake vortex elements to the velocity in the rotor plane is negligible. Deleting these elements helps speeding up
the calculations. Figure 2 shows the combination of the two implemented wake coarsening methods. The wake coarsening methods are a function of rotor revolutions. Because the effect of the vortex elements on the induced velocity is a function of the distance, the parameters for these methods will be dependent on the wind speed. The latter has an impact on the rotor speed and on the convection speed of the vortex elements. The wake coarsening parameters that we used for our simulations are given in table A1.





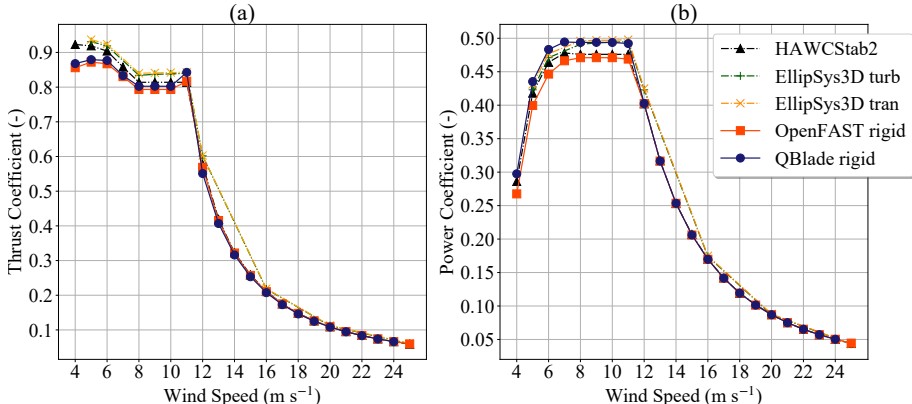

**Figure 3.** Performance coefficients for aerodynamic simulations with idealized conditions: (a) Thrust coefficient; (b) Power coefficient.

Regarding the BEM simulations, we did not include the wake memory effect model in our simulations. The main reason for this is that this model was not present in older FAST versions (e.g. FAST V8) and has been introduced fairly recently in OpenFAST. Therefore, we haven't sufficiently tested this new module to be confident in its results. For this study, it can be considered that the aerodynamic model of the BEM simulations is comparable to the one used in the older FAST V8 code.

## 3  Baseline Comparison and Performance under Idealized Conditions

To do a baseline comparison of our aerodynamic models, we ran a series of idealized aerodynamic simulations. The parameters for these simulations are summarized in Table 2 under the column 'Aerodynamic calculations'. With these settings the flow is axis-symmetric on the rotor and no elasticity is taken into account. Under these conditions, the differences between the aerodynamic models are minimized as far as possible. Table 1 shows that under these conditions the only differences between the methods are the tip- and root-loss model and the turbulent wake state for high tip speed ratios (i.e. low wind speeds). The total simulation time in these conditions is 400 s for wind speeds below 12 m/s and 300 s for wind speeds of 12 m/s and higher. For the LLFVW simulations, the turbine reaches a steady state after about 200 s for wind speeds below 12 m/s and after about 100 s for wind speeds of 12 m/s and above. We used the averaged values of the of the last 30 s of simulation time for the comparisons below. These simulations include the interaction with the turbine controller.

Figure 3 shows the performance coefficients for aerodynamic calculations when done with the BEM and LLFVW codes. In general, the performance coefficients from both calculations agree well. The thrust coefficient from LLFVW calculations follows the thrust coefficient from BEM calculations very closely (Fig. 3 (a) ). It is only at a wind speed of 11 m/s that the values visibly differ. As for the power coefficient (Fig. 3 (b) ), the LLFVW code predicts higher values for wind speeds below rated wind speed. Above rated wind speed, the power coefficients in both codes almost perfectly match. This behavior can be explained from the fact that at higher wind speeds, the turbine controller pitches the blades out to keep the power output of the turbine constant. The controller logic is identical in both codes. Additionally, at higher wind speeds the rotor speed is





**Table 2.** Simulation parameters for aerodynamic and aeroelastic simulations

| Parameter | Aerodynamic calculations | Aeroelastic calculations | |
| --- | --- | --- | --- |
| | | Sensitivity study | Turb. calculations |
| Mean $V_{Hub}$ | 4 - 25 m/s | 4 - 24 m/s | |
| Wind model | constant uniform | constant uniform | IEC NTM |
| Elasticity | off | on | |
| Rotor cone / Shaft tilt angles | 0° / 0° | 2.5° / 5° | |
| Wind shear exponent | 0 | 0 | 0.2 |
| Upflow angle | 0° | 0° | 8° |
| Nacelle yaw angle | 0° | 0° | -8°, 0°, 8° |
| Wake coarsening | | See Table A1 | |
| Rotor azimuth step / Time step | 5° | 0.04 s | |

kept constant by the controller while the convection speed of the wake increases. This decreases the influence of the wake on the turbine's thrust and power and hence the differences in the aerodynamic models become smaller. If we compare numerical values at 8 m/s, the difference between the thrust and power coefficients from both codes is 1.1% and 4.6% respectively. Similar differences of power and thrust between BEM and LLFVW codes for 8 m/s and ideal inflow conditions were also reported in

(Madsen et al., 2012).

Figure 3 also contains data from three calculations done with other codes. The data is taken from (Bak et al., 2013), where the performance coefficients of the rigid DTU 10 MW RWT are calculated with the BEM-based code HAWCStab2 and the CFD-based code EllipSys3D. For the latter, two different boundary layer models were used. The OpenFAST and HAWCStab2 calculations predict very similar performance coefficients except for low wind speeds. QBlade predicts thrust coefficients that

are closer to the BEM-based codes and power coefficients that are closer to the CFD-based codes.

The turbulent load calculations described in Sect. 4 used the full aeroelastic turbine model. The simulation parameters for the full aeroelastic model are summarized in Table 2 under the column 'Aeroelastic calculations'. Because of the long simulation time of each load case, we applied more aggressive wake coarsening parameters for the aeroelastic calculations than for the aerodynamic calculations. These are also summarized in Table A1. These simulation parameters are the result of a sensitivity

study we performed to make sure that our chosen, wind dependent, wake parameters for the aeroelastic LLFVW simulations predicted similar steady state values compared to the idealized aerodynamic calculations with long wakes.

Figure 4 shows the comparison of the rotor thrust, rotor power, pitch angle and rotor speed for the aerodynamic and aeroelastic calculations. Using the parameters in Table 2 has only a small influence on the steady state values of the rotor thrust and power (Figs. 4 (a) and 4 (b) ). For both OpenFAST and QBlade, the rotor thrust from aeroelastic calculations is slightly higher

that the thrust from purely aerodynamic calculations. This comes from the coned rotor used in the aeroelastic calculations. Coning the rotor changes the relative direction of the centrifugal force so that it has a component normal to the rotor plane.



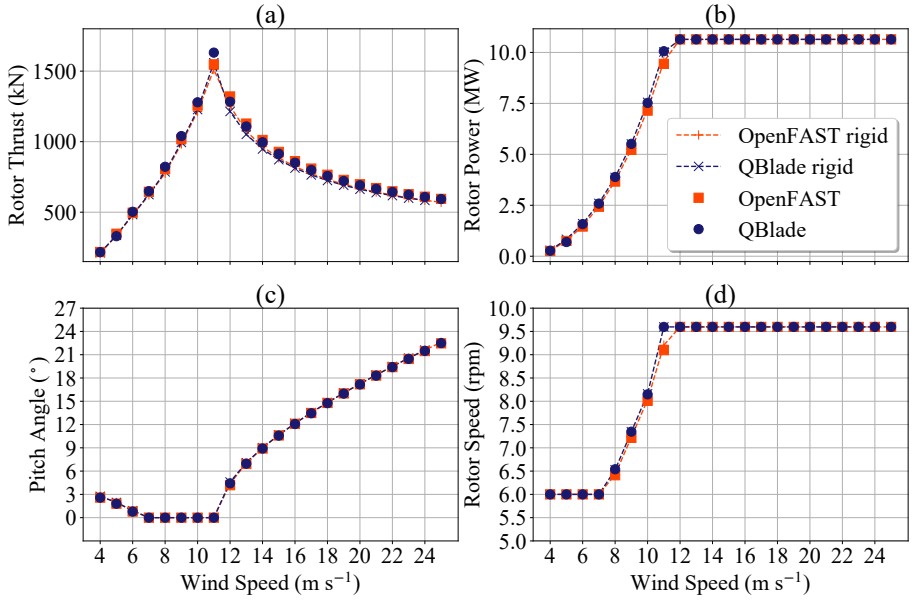

**Figure 4.** Comparison of aerodynamic and aeroelastic calculations on turbine performance: (a) Rotor thrust; (b) Rotor power; (c) Pitch angle; (d) Rotor speed.

The rotor speeds and pitch angles for the aerodynamic and aeroelastic calculations are shown in Figs. 4 (c) and 4 (d) respectively. In these subfigures we can see that there is also little difference in the controller signals if the turbine is simulated aeroelastically. The pitch angle coincides for all simulations. As for the rotor speed, QBlade predicts higher rotor speeds than OpenFAST for wind speeds between 7 and 11 m/s. Particularly for 11 m/s wind speed, QBlade simulations already reach the rated rotor speed while OpenFAST predict a steady state rotor speed of 9.1 rpm. This fact explains the higher thrust (Fig. 4 (a) ) and thrust coefficient (Fig. 3 (a) ) for this wind speed.

An important result from Fig. 4 is that using the wake coarsening parameters from Table A1 barely affects the accuracy of the aeroelastic steady state results compared to the aerodynamic results. Therefore, the coarsening parameters can be used to speed up the turbulent load calculations in the next section.

## 4 Design Load Calculations with Turbulent Wind

The turbulent wind load cases were performed following the DLC groups 1.1/1.2 from the IEC61400-1 standard (IEC 61400-1 Ed. 3). The turbine setup for these load cases is listed in Table 2 in the third column. In this study, we considered wind speed bins (defined by the mean $V_{Hub}$) between 4 m/s to 24 m/s in 2 m/s steps. For each wind speed bin, six simulations were performed using two turbulence seeds per yaw angle. The same wind fields were used for BEM and LLFVW calculations. In total we did 66 simulations with 600 s simulation time for both the BEM and LLFVW codes. To give time for the wake to develop in the LLFVW calculations, we included an extra 100 s simulation time that was discarded in the load analysis. These



**Table 3.** Considered sensors and analysis type for turbulent load calculations. C.S. = Coordinate System; F = Fatigue; U = Ultimate.

| Sensor Name | OpenFAST Coord. Sys. | Symbol | Analysis Type |
|---|---|---|---|
| Blade root in-plane / out-of-plane bending moment | Coned C.S. **c** | $M_X^{\mathrm{BR}}$ / $M_Y^{\mathrm{BR}}$ | F / U |
| Yaw bearing roll / tilt / yaw moment | Nacelle C.S. **n** | $M_X^{\mathrm{YB}}$ / $M_Y^{\mathrm{YB}}$ / $M_Z^{\mathrm{YB}}$ | F / U |
| Tower base side-side / fore-aft / torsional bending moment | Tower-base C.S. **t** | $M_X^{\mathrm{TB}}$ / $M_Y^{\mathrm{TB}}$ / $M_Z^{\mathrm{TB}}$ | F / U |
| Blade tip out-of-plane / in-plane deflection | Coned C.S. **c** | $D_X^{\mathrm{BT}}$ / $D_Y^{\mathrm{BT}}$ | U |
| Tower top fore-aft / side-side deflection | Tower-top C.S. **p** | $D_X^{\mathrm{TT}}$ / $D_Y^{\mathrm{TT}}$ | U |
| Blade pitch angle / Rotor speed | N.A. | $\theta$ / $\Omega$ | F / U |

discarded 100 s wake build-up time were also included in the BEM-simulations to make sure that for both codes we had the same incoming wind conditions.

### 4.1 Considered Sensors

For the analysis of the turbulent wind load calculations, we considered a selection of load sensors that is representative of the
dynamics and load level of the entire turbine. The sensors include the blade root bending moments, the yaw bearing moments and the tower base bending moments. In addition, we considered the blade tip and tower top deflections. As for the controller signals, we analyzed the collective pitch angle and the rotor speed. Table 3 lists all considered sensors for this study and their corresponding symbol. For each sensor group, we used the coordinate systems defined in (Jonkman and Buhl, 2005) for both OpenFAST and QBlade calculations. The coordinate systems are listed in Tab. 3. In addition, the table also lists the type of
post-processing analysis that we performed for each sensor group. F stands for fatigue load analysis and U for ultimate load analysis.

### 4.2 Statistical Overview

Figure 5 shows an overview of the statistical values of rotor thrust, electrical power, pitch angle and rotor speed for the turbulent wind calculations of both codes. The markers joined with lines represent the median of all six 600 s simulations in each wind
bin. The shaded area represents the Inter-Quartile Range (IQR) – the range in which 50% of the simulation values lie. The error bars represent the extrema of all values recorded at one wind speed bin.

    Let us consider rotor thrust in Fig. 5 (a) first. We can see that for wind speeds lower than the rated wind speed the values of the rotor thrust calculated with the LLFVW code tend to be higher than the values from the BEM code. This tendency inverts for wind speed between 12 and 18 m/s. Here, the medians and IQRs from the BEM calculations are slightly higher that the cor-
responding values from LLFVW calculations. For wind speeds higher than 18 m/s, the medians and IQRs between both codes almost match. This behavior of the thrust as a function of the wind speed is also seen for the steady state values in Fig. 4 (a). When we consider the extrema of the rotor thrust, we can see that for almost all wind speeds, the ranges between the medians



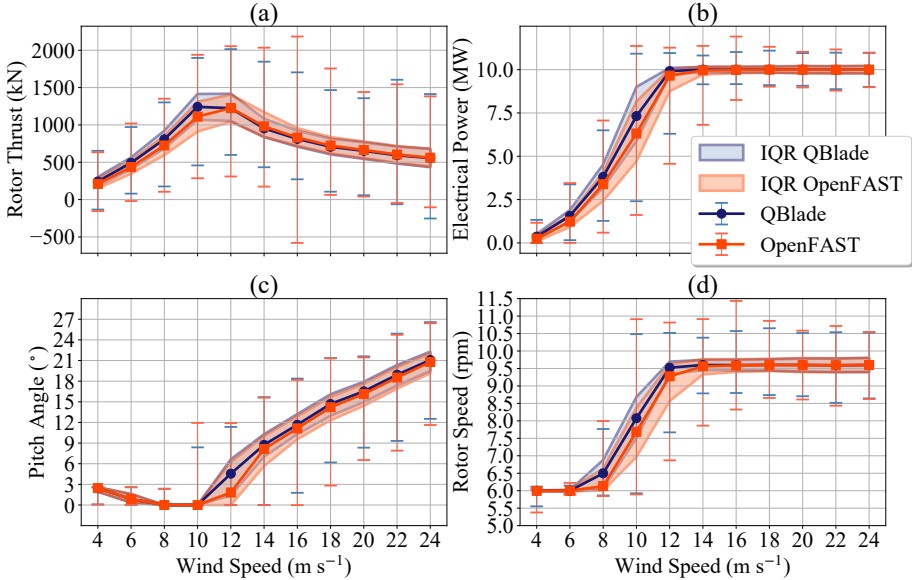

**Figure 5.** Comparison of statistical values for turbulent calculations: (a) Rotor thrust; (b) Electrical power; (c) Pitch angle; (d) Rotor speed. IQR = Inter-quartile range.

and the extrema from the LLFVW calculations are smaller than the median-extrema ranges from the BEM calculations. The exceptions are the wind bins of 4, 22 and 24 m/s. This leads us to expect higher extrema in the out-of-plane turbine loads.

The comparison of electrical power from the turbulent wind simulations – Fig. 5 (b) – also shows similarities with the comparison in ideal situations (Fig. 4 (b) ). For below rated wind speeds, the LLFVW simulations show higher medians of the

electrical power than results from BEM simulations. In contrast, the IQRs are lower for the LLFVW simulations. For the 12 m/s wind speed bin, the median electrical power for the LLFVW calculation is already practically 10 MW while the median power of the BEM simulations is still 9.6 MW. Also, we can see from the IQRs that at 12 m/s wind speed a higher percentage of time the power from BEM lies below rated power. Regarding the extrema, the simulations with the LLFVW code show almost always lower range between extrema and medians than the BEM simulations. The only exception occurring at 4 m/s.

The rotor speed signal (Fig. 5 (d) ) is closely linked to the power signal (Fig. 5 (b) ). The most observations made for the electrical power also hold true for the rotor speed. The exception being the IQRs of the signal for the 8 m/s wind bin. Here the IQR of the rotor speed is smaller in the BEM simulations than in the LLFVW simulations.

Finally, Fig. 5 (c) shows a comparison of the pitch angles between both codes. For wind speeds between 4 m/s and 8 m/s, there is practically no difference between the statistical values from the BEM and the LLFVW simulations. For higher wind

speeds, we can see that the LLFVW simulations have slightly higher median, first quartile and third quartile values than BEM simulations. This behavior was not seen in the idealized aeroelastic simulations – Fig. 4 (c) – where the steady state values of the pitch angles where almost identical. For each wind speed bin, the range between the median and extrema of the pitch angles is larger in the BEM simulations compared to the LLFVW simulations. For the wind speed bins of 16, 20, 22 and 24 m/s, the



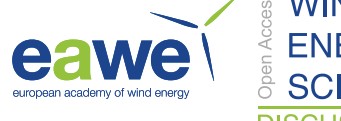

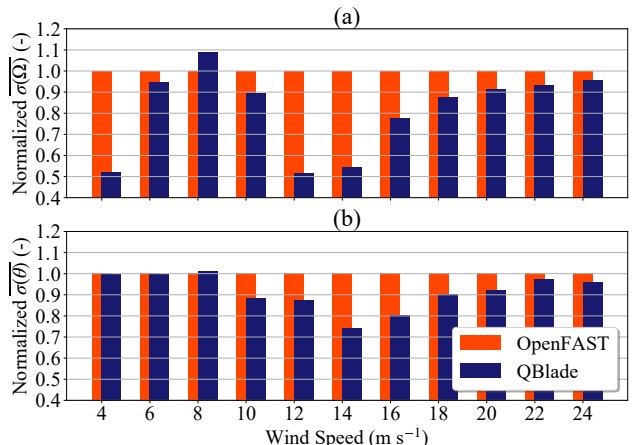

**Figure 6.** Normalized averaged standard deviations vs. wind speed. (a) Rotor speed $\Omega$; (b) Pitch angle $\theta$

maxima of the pitch angles are higher for the LLFVW simulations compared to the BEM simulations. This can be explained by the generally higher average values of the pitch angle in the former.

Differences in the IQRs shown in Fig. 5 reveal that the variability of the signals changes if we use different aerodynamic models. Particularly in the wind speeds from 10 m/s to 14 m/s, the IQR of the pitch angle, the rotor speed and the electrical

power from the BEM calculations is visibly larger than the IQR of these signals from the LLFVW calculations. A quantitative analysis of these variations and their effect on the loads is done in the subsequent sections.

## 5   Fatigue Analysis of the Design Load Calculation Results

In this section, we discuss the influence of the aerodynamic models on the variability of control signals and fatigue loads of different turbine sensors. The analysis is based on the results of the turbulent load calculations described in the previous

section. In this and the following sections, the subscripts $(\cdot)_{\text{BEM}}$ and $(\cdot)_{\text{LLFVW}}$ denote values obtained from BEM and LLFVW simulations, respectively.

### 5.1   Controller Signals

To quantify the variability of the control signals, we used the standard deviation $\sigma(\cdot)$ as our metric. For each of the six simulations in one wind speed bin, we calculate $\sigma$ for the rotor speed $\Omega$ and the pitch angle $\theta$. We then average all six standard

deviations for each control signal to get a representative quantity for the signal's variability for that wind speed bin. These averaged standard deviations are denoted $\overline{\sigma(\theta)}$ for the pitch angle and $\overline{\sigma(\Omega)}$ for the rotor speed.

Figure 6 shows the normalized $\overline{\sigma(\theta)}$ and $\overline{\sigma(\Omega)}$ for the all the simulated wind speed bins. The normalization is with respect to the values from the BEM simulations, so the normalized $\overline{\sigma(\theta)}_{\text{BEM}}$ and $\overline{\sigma(\Omega)}_{\text{BEM}}$ are always 1.



If we consider the rotor speed (Fig. 6 (a) ), we see that for all wind speed bins except 8 m/s, the normalized $\overline{\sigma(\Omega)}_{\text{LLFVW}}$ is lower than 1. The largest deviations can be seen at wind speed bins of 4, 12 and 14 m/s. Here, the normalized $\overline{\sigma(\Omega)}_{\text{LLFVW}}$ is almost 0.5. When we consider wind speed bins of 16 m/s and above, we see the normalized value of $\overline{\sigma(\Omega)}_{\text{LLFVW}}$ increase monotonically towards 1.

5     Why do we have these differences in the wind speed bins 4 m/s, 8 m/s, 12 m/s and 14 m/s? In the case of the 4 m/s wind bin, this difference can be explained if we look at Fig. 5 (d). For very low wind speeds, the value of $\Omega$ is almost always $\Omega_{min}$. Yet in the BEM simulations, there are load cases where $\Omega_{\text{BEM}}$ drops below $\Omega_{min}$ and reaches a lower value than $\Omega_{\text{LLFVW}}$. Because of the small absolute value of $\overline{\sigma(\Omega)}$ at those wind speeds, those excursions of $\Omega_{\text{BEM}}$ have a higher relative effect on $\overline{\sigma(\Omega)}_{\text{BEM}}$ and also on the normalized $\overline{\sigma(\Omega)}_{\text{LLFVW}}$.

10     As for $\overline{\sigma(\Omega)}$ at wind speed bins of 12 and 14 m/s, the large differences come from the missing wake memory effect in the BEM calculations. An analysis explaining this phenomenon is presented in section 5.3.

The fact that for the 8 m/s wind speed bin the normalized $\overline{\sigma(\Omega)}_{\text{LLFVW}}$ is 1.09 can be explained if we look again at Fig. 5 (d). The median of $\Omega_{\text{LLFVW}}$ is 6.5 rpm while in the BEM simulations, the median $\Omega_{\text{BEM}}$ is 6.1 rpm. Because the torque controller keeps $\Omega$ equal or above $\Omega_{min}$, the number of occurrences in this wind speed bin where $\Omega = \Omega_{min}$ is higher in the BEM simulations compared to the LLFVW simulations. Hence, the total variability of the signal will be lower leading to a smaller $\overline{\sigma(\Omega)}_{\text{BEM}}$ compared to $\overline{\sigma(\Omega)}_{\text{LLFVW}}$.

For the pitch angle signal we can see that the normalized $\overline{\sigma(\theta)}_{\text{LLFVW}}$ behaves differently as a function of wind speed than $\overline{\sigma(\Omega)}_{\text{LLFVW}}$ (Fig. 6 (b) ). For wind speed bins between 10 and 16 m/s, $\overline{\sigma(\theta)}_{\text{LLFVW}}$ drops to values significantly lower than 1, reaching a value of 0.74 for the 14 m/s wind speed bin. For low wind speed bins, $\overline{\sigma(\theta)}_{\text{LLFVW}}$ is practically 1. For the 18 and 20 20   m/s wind speed bins, $\overline{\sigma(\theta)}_{\text{LLFVW}}$ is above 0.9 and for 22 and 24 m/s wind bins, the normalized standard deviations are above 0.95. The low values of $\overline{\sigma(\theta)}_{\text{LLFVW}}$ at wind speeds around rated wind speed is also due to the missing wake memory effect in BEM calculations and will be analyzed in section 5.3.

## 5.2   Loads

The fatigue loads are quantified using the Damage Equivalent Loads (DELs) metric. DELs are derived from the time series of 25   the load sensor using a rain-flow counting algorithm. In this algorithm, the time-varying signal is broken down into individual cycles by matching local minima and maxima in the time series (Hayman, 2012). The rain-flow counting was performed using NREL's post-processing software Crunch (Buhl). We used the Palmgren-Miner linear damage accumulation hypothesis to obtain the DELs. Two types of fatigue loads were calculated. The first type are the short-term 1 Hz DELs – noted $\text{DEL}_{1\text{Hz}}(\cdot)$ – which give us the equivalent fatigue damage of one simulation. The second type are the lifetime DELs – noted $\text{DEL}_{\text{Life}}(\cdot)$ 30   – which give us the equivalent loading for the entire turbine lifetime. The lifetime DELs were obtained following the method described in (Hayman, 2012). We used the wind distribution corresponding to wind class IA turbine with 20 years design life and an equivalent cycle number of $10^7$. For the blade root fatigue loads, we used an inverse S-N curve-slope of $m = 10$ to calculate the DELs. For all other loads, the inverse S-N curve-slope used is $m = 4$.





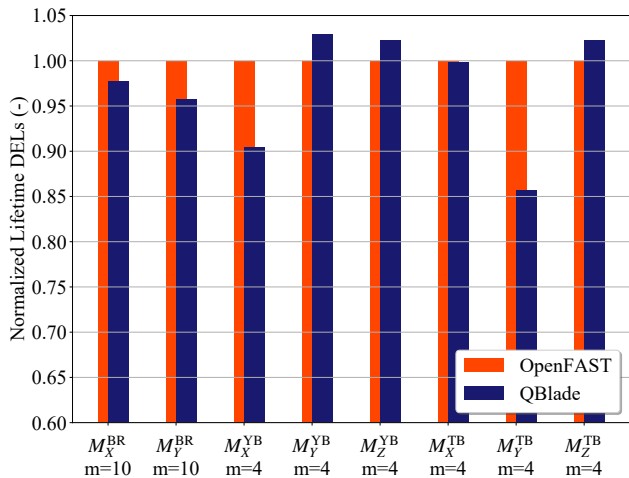

**Figure 7.** Normalized lifetime DELs for the considered turbine load sensors. Sensor notation is given in Tab. 3.

Figure 7 shows the normalized lifetime DELs for the considered turbine load sensors. We can see from this figure that performing the simulations with different aerodynamic models has an impact on the lifetime DELs of almost all considered load sensors. Let us start with the blade root. For these loads we see that the normalized $\mathrm{DEL}_{\mathrm{Life}}(M_X^{\mathrm{BR}})_{\mathrm{LLFVW}}$ and $\mathrm{DEL}_{\mathrm{Life}}(M_Y^{\mathrm{BR}})_{\mathrm{LLFVW}}$ are 0.98 and 0.96, respectively. The finding that the fatigue loads of $M_{Y-\mathrm{LLFVW}}^{\mathrm{BR}}$ are lower than $M_{Y-\mathrm{BEM}}^{\mathrm{BR}}$

has also been reported by other studies, e.g. (Perez-Becker et al., 2018; Hauptmann et al., 2014; Boorsma et al., 2016).

When considering the yaw bearing, Fig. 7 shows that the normalized $\mathrm{DEL}_{\mathrm{Life}}(M_X^{\mathrm{YB}})_{\mathrm{LLFVW}}$ has an even lower value than the blade root fatigue loads: 0.91. If we look at the other bending moments, we see the opposite behavior: simulations with the LLFVW code predict higher fatigue loads for these sensors. The normalized values of $\mathrm{DEL}_{\mathrm{Life}}(M_Y^{\mathrm{YB}})_{\mathrm{LLFVW}}$ and $\mathrm{DEL}_{\mathrm{Life}}(M_Z^{\mathrm{YB}})_{\mathrm{LLFVW}}$ are 1.04 and 1.02, respectively. The largest difference in the lifetime DELs of all considered sensors

occurs in the tower base fore-aft bending moment. The normalized value of $\mathrm{DEL}_{\mathrm{Life}}(M_Y^{\mathrm{TB}})_{\mathrm{LLFVW}}$ is 0.86. In contrast, the normalized value of $\mathrm{DEL}_{\mathrm{Life}}(M_X^{\mathrm{TB}})_{\mathrm{LLFVW}}$ is practically 1.

When calculating the lifetime fatigue loads, we take into account the loading of all the wind speed bins. In different wind speed bins, the turbine can see qualitatively different loading scenarios leading to significant differences in fatigue loading when simulated with different aerodynamic models. To further understand which phenomena are contributing to the differences in

fatigue loads, we also analyzed the contribution of the individual wind speed bins to the fatigue loading of the sensors. As we can see in Fig. 8, the contribution of the wind speed bins to the lifetime fatigue loads is strongly dependent on the wind speed. To limit the extent of the fatigue analysis, we will concentrate on the load sensors that show the largest differences in lifetime DELs: $M_Y^{\mathrm{BR}}$, $M_X^{\mathrm{YB}}$, $M_Y^{\mathrm{YB}}$ and $M_Y^{\mathrm{TB}}$.

Figure 8 (a) shows the normalized average 1 Hz DEL for $M_Y^{\mathrm{BR}}$ as a function of the wind speed bin. The average, noted

$\overline{\mathrm{DEL}}_{1\mathrm{Hz}}(\cdot)$, was taken from the 1Hz DELs of each of the 6 realizations in one wind speed bin. The normalization was done with respect to the values of the BEM simulations. We can see in this subfigure that the value of the normalized $\overline{\mathrm{DEL}}_{1\mathrm{Hz}}(M_Y^{\mathrm{BR}})_{\mathrm{LLFVW}}$

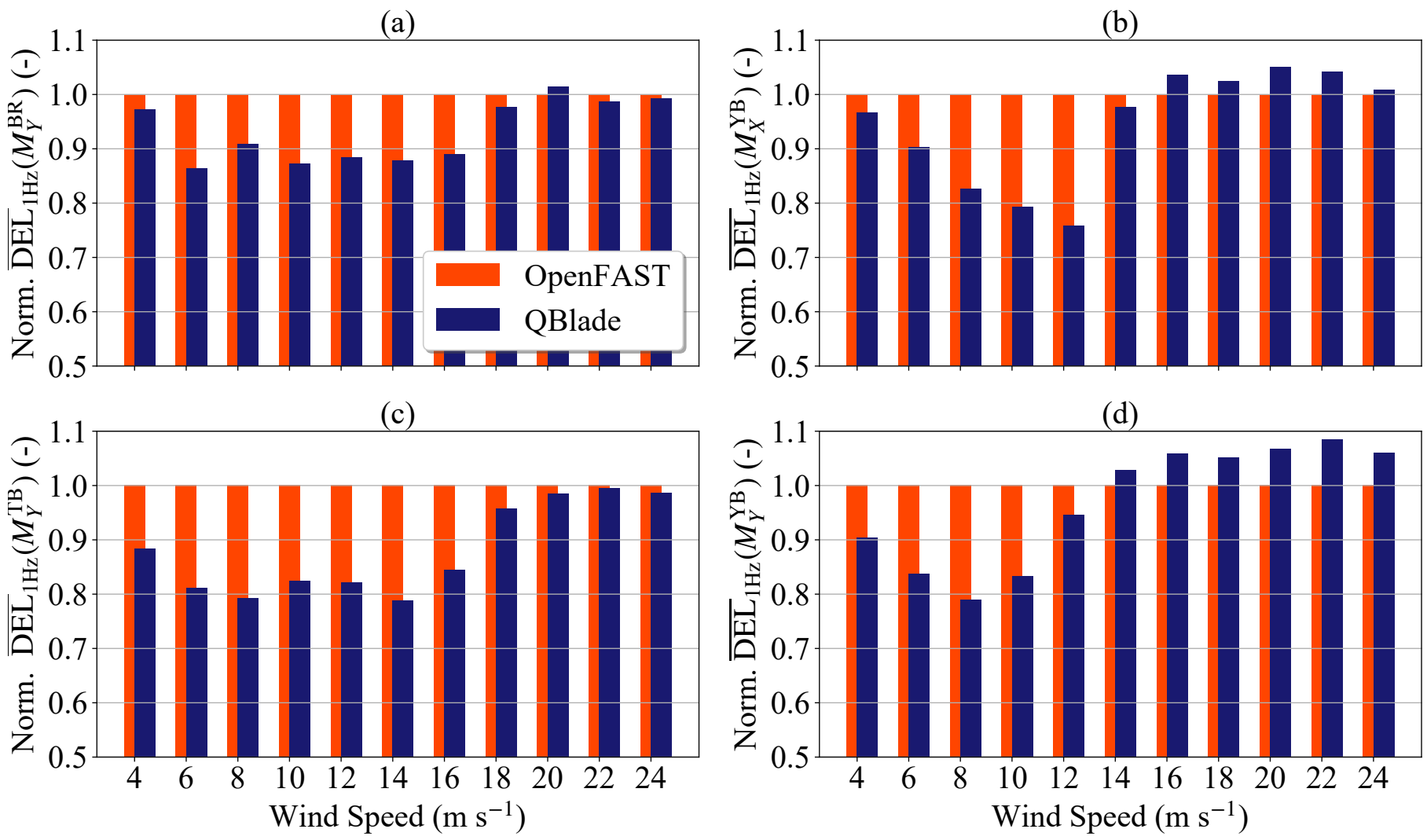

**Figure 8.** Normalized averaged 1 Hz DEL as a function of the wind speed bin. (a) Blade root out-of-plane bending moment $M_Y^{\mathrm{BR}}$; (b) Yaw bearing roll moment $M_X^{\mathrm{YB}}$; (c) Tower base fore-aft bending moment $M_Y^{\mathrm{TB}}$; (d) Yaw bearing tilt moment $M_Y^{\mathrm{YB}}$.

is lower than 1 for all wind speed bins except 20 m/s, where the value is 1.014. Yet there are differences. It is for wind speed bins between 6 and 16 m/s that the values of the normalized $\overline{\mathrm{DEL}}_{\mathrm{1Hz}}(M_Y^{\mathrm{BR}})_{\mathrm{LLFVW}}$ are the lowest: they take values around and below 0.9. For the wind speed bins 18 m/s, 22 m/s and 24 m/s , the normalized values of $\overline{\mathrm{DEL}}_{\mathrm{1Hz}}(M_Y^{\mathrm{BR}})_{\mathrm{LLFVW}}$ are almost 1.

A qualitative similar behavior can be seen for the $M_Y^{\mathrm{TB}}$ sensor in Fig. 8 (c). For wind speeds bins of up to 16 m/s, the effects of using different aerodynamic modules can clearly be seen, as the values of $\overline{\mathrm{DEL}}_{\mathrm{1Hz}}(M_Y^{\mathrm{TB}})_{\mathrm{LLFVW}}$ lie in a range between 0.79 and 0.89. If we consider higher wind speeds, the differences become smaller: for wind speed bins of 18 m/s and higher, the values of the normalized $\overline{\mathrm{DEL}}_{\mathrm{1Hz}}(M_Y^{\mathrm{TB}})_{\mathrm{LLFVW}}$ are 0.96 and higher.

The behavior of the fatigue damage on the yaw bearing sensors is qualitatively different from the tower base fore-aft- and the blade root out-of-plane bending moments. Figure 8 (b) shows the values of $\overline{\mathrm{DEL}}_{\mathrm{1Hz}}(M_X^{\mathrm{YB}})_{\mathrm{LLFVW}}$ for all the simulated wind speed bins. The normalized values lie well below 1 for all wind speed bins up to 12 m/s, reaching a minimum of 0.76 at 12 m/s wind speed. For wind speed bins of 14 m/s and above, $\overline{\mathrm{DEL}}_{\mathrm{1Hz}}(M_X^{\mathrm{YB}})_{\mathrm{LLFVW}}$ rises sharply reaching values higher than 1 for wind speed bins of 16 m/s and above. The highest value is of $\overline{\mathrm{DEL}}_{\mathrm{1Hz}}(M_X^{\mathrm{YB}})_{\mathrm{LLFVW}}$ is 1.05 and is reached at the wind speed bin of 20 m/s.

If we consider the yaw bearing tilt moment, the behavior is again different from the other considered sensors (Fig. 8 (d) ). At wind speed bins between 4 and 12 m/s, the values of $\overline{\mathrm{DEL}}_{\mathrm{1Hz}}(M_Y^{\mathrm{YB}})_{\mathrm{LLFVW}}$ lie below 1. The minimum of this range occurs at a wind speed bin of 8 m/s and has a value of 0.79. As the wind speed increases, the trend inverses and the normalized values of $\overline{\mathrm{DEL}}_{\mathrm{1Hz}}(M_Y^{\mathrm{YB}})_{\mathrm{LLFVW}}$ take values above 1. The maximum of 1.09 is reached at the wind speed bin of 22 m/s. This distribution of the short term fatigue loads also explains why the normalized $\mathrm{DEL}_{\mathrm{Life}}(M_Y^{\mathrm{YB}})_{\mathrm{LLFVW}}$ is 1.04 in Fig. 7.



## 5.3 Discussion

To better understand the differences in the fatigue loads and the variability of the controller signals, we can categorize the wind speed bins into three qualitatively different wind speed regions: Regions A, B and C.

– Region A includes wind speed bins between 4 and 10 m/s. In this region, the turbine is below rated wind speed and hence the controller seeks to maximize energy capture. The pitch controller is largely inactive and the tip speed ratio of the turbine is above or close to the turbine's optimal tip speed ratio. For the aerodynamic loads this means that the axial induction factor is relatively large. Therefore, the differences in the aerodynamic modeling will be large and their influence on the turbine loads significant.

– Region B encompasses wind speed bins between 10 m/s and 16 m/s. In this region, the transition between below-rated power and above-rated power operations of the controller occurs. Small differences in aerodynamic loads can trigger this transition and significantly affect the turbine loading. This is because around rated wind the thrust on the rotor is highest (Fig. 4 (a) ) and the activation of the pitch controller influences the thrust considerably. In this region, the tip speed ratio of the turbine is still close to the optimum. Hence the axial induction is still large making differences in aerodynamic models relevant for turbine loading.

– Region C covers wind speed bins between 18 m/s and 24 m/s. Here, the blade pitch angle is relatively high and the rotor speed is close to the rated rotor speed $\Omega_R$. With higher wind speeds the wake is convected faster downstream, effectively reducing the effect of its induced velocity on the rotor plane. This, in addition to the high pitch angles of the blade, leads to smaller contribution of the axial induced velocities to the total relative velocity seen by the blades in the rotor plane. Hence, effect of the aerodynamic models on the controller behavior and loads decreases.

Because we are analyzing turbulent load calculations with varying wind speed, the limits between the regions cannot be exactly defined. We will consider one wind speed bin for each region as a representative set of simulations for that region. For each chosen wind speed bin, the qualitative turbine behavior will be the same as in the corresponding region described above. For Region A the chosen wind speed bin is 8 m/s, for Region B the wind speed bin is 14 m/s and for Region C, the wind speed bin is 20 m/s. We will concentrate on the same turbine loads as in Fig. 8 since they showed the highest differences in lifetime fatigue loads.

### 5.3.1 Blade Root Out-of-Plane Bending Moment

Figure 9 shows the Power Spectral Density (PSD) plots of $M_Y^{\mathrm{BR}}$ for several BEM and LLFVW simulations. Each row of the subplots in the figure corresponds to one of the aforementioned regions. The left column shows PSD plots of the results from turbulent wind load calculation. The right column shows the PSDs of 200 s simulations with steady inflow conditions. The latter column will help us understand the source of the differences in the fatigue loads between both codes. As with the turbulent calculations, additional 100 s were simulated and discarded in the analysis to allow the wake in the steady LLFVW simulations to build up.



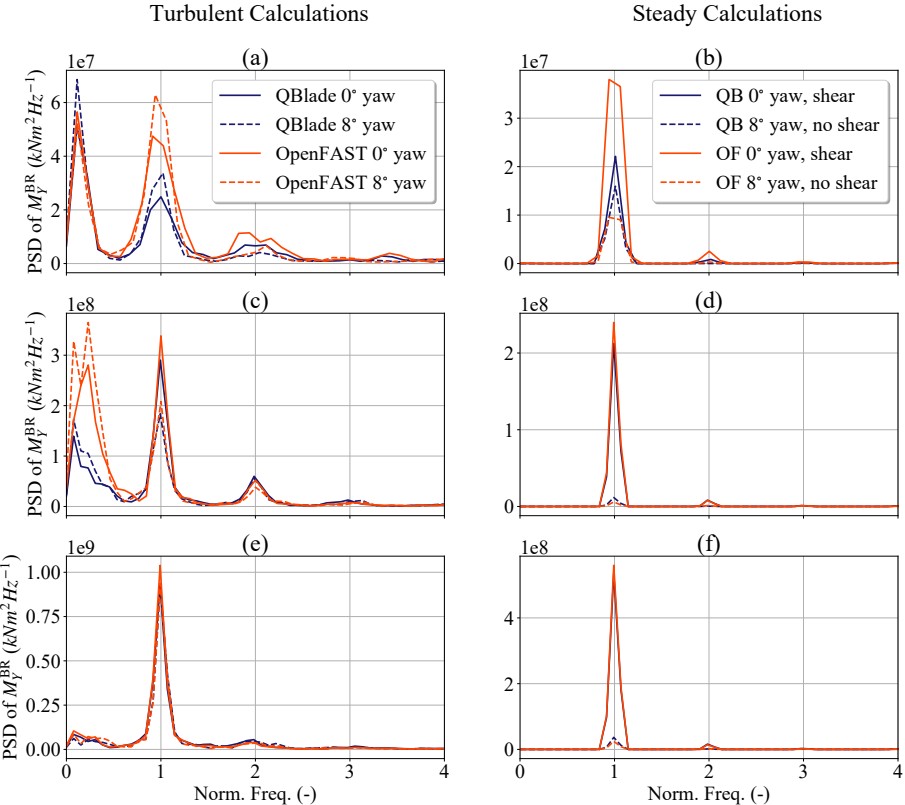

**Figure 9.** Power Spectral Density plots for $M_Y^{\mathrm{BR}}$ at different wind speeds. (a) Turbulent calculations at 8 m/s wind speed; (b) Steady calculations at 8 m/s wind speed; (c) Turbulent calculations at 14 m/s wind speed; (d) Steady calculations at 14 m/s wind speed; (e) Turbulent calculations at 20 m/s wind speed; (f) Steady calculations at 20 m/s wind speed

Figures 9 (a) and 9 (b) show the PSD plots of $M_Y^{\mathrm{BR}}$ for simulations in the 8 m/s wind speed bin – i.e. Region A. In Fig. 9 (a) the solid lines represent turbulent wind simulations with a $0°$ yaw error while the dashed lines represent simulations with $8°$ yaw error. For Fig. 9 (b), the solid lines represent results from steady wind simulations without yaw error but with a 0.2 wind shear exponent while the dashed lines represent results from simulations with $8°$ yaw error and a wind shear exponent of 0.

5   The idea of the simulations in Fig. 9 (b) is to isolate different aerodynamic phenomena to see their individual contribution to the fatigue loading. Apart from the tip- and root-loss model, the major difference of the aerodynamic models in the solid line simulations is the treatment of the non-homogeneous wind speed distribution on the rotor disk. In contrast, the major difference of the aerodynamic models in the dashed line simulations is the treatment of the oblique inflow.

When we consider the PSDs of the turbulent load calculations (Fig. 9 (a) ) we can see that the main differences between

10   the PSDs of the BEM and the LLFVW simulations occur at the once-per-revolution or 1P frequency. Within each code, the amplitude of the PSD at the 1P frequency is higher in the $8°$ yaw simulations than in the $0°$ yaw simulations. This is true for both the BEM and the LLFVW simulations. Two of the main contributions to the 1P loading of $M_Y^{\mathrm{BR}}$ are the wind shear and





the yaw misalignment of the rotor. Because both are present in the $8°$ yaw simulations, the total variation of the loads at that frequency will be higher.

If we now compare the amplitude between the aerodynamic codes we can see that, for both the $8°$ and $0°$ yaw error simulations, the amplitude of the 1P peak in the PSD of the BEM simulations is larger than the corresponding peak in the LLFVW

simulations. The main source of this difference between both codes is the effect that the non-homogenous wind field –arising from the wind shear– has on the local blade aerodynamics. As Fig. 9 (b) shows, simulating the turbine in sheared inflow leads to the largest differences between both codes in the load prediction at the 1P frequency of PSD($M_Y^{\mathrm{BR}}$). The reason for this difference has already been identified and explained by other authors – e.g. (Madsen et al., 2012; Boorsma et al., 2016) – and will only be briefly mentioned here. According to (Moriarty and Hansen, 2005), AeroDyn calculates the local thrust coefficient

using the average inflow wind speed from the rotor; a procedure also done in other BEM codes (Madsen et al., 2012). This choice has an averaging effect on the local axial induced velocity when the turbine is simulated with sheared inflow. As a result, the local angle of attack sees a higher amplitude in the 1P variations compared to when the scenario simulated with a LLFVW code. In the latter, the local three-dimensional induction field is implicitly modelled through the lifting line and the induced velocities from the wake vortices. The result is a better tracking of the local axial induced velocity with the LLFVW

simulations. Having higher angle of attack variations in BEM simulations leads to higher 1P variations in the local lift forces and ultimately to higher 1P variations in $M_{Y-\mathrm{BEM}}^{\mathrm{BR}}$ (compared to $M_{Y-\mathrm{LLFVW}}^{\mathrm{BR}}$).

The qualitative behavior changes when we compare simulations in Region B (Figs. 9 (c) and 9 (d) ). While there are still some signal differences at the 1P frequency of the PSD, the main differences between both codes now occur in the low frequency region of the PSD. This is the frequency region where the controller is active. If we recall the differences in the controller

signals from Fig. 6, it is also in Region B that the largest differences in the normalized $\overline{\sigma(\theta)}_{\mathrm{LLFVW}}$ and $\overline{\sigma(\Omega)}_{\mathrm{LLFVW}}$ lie. In contrast, Fig. 9 (d) shows the PSDs of the steady calculations where there is minimal controller action. The differences in PSD($M_Y^{\mathrm{BR}}$) there are relatively small. Hence, a qualitative change in the controller behavior is causing the large differences in PSD($M_Y^{\mathrm{BR}}$) in Region B. Because both codes differ only in their aerodynamic models, this difference in the controller behavior must ultimately have its origin in the different aerodynamic implementations.

Further insight can be gained from Figure 10, where the time series of the controller signals from simulations with 14 m/s average hub wind speed are shown. We can see in Figs. 10 (a) and (b) that while subjected to the same turbulent wind field, $\Omega_{\mathrm{BEM}}$ varies significantly more than $\Omega_{\mathrm{LLFVW}}$. As a result, the oscillations of $\theta_{\mathrm{BEM}}$ have a higher amplitude which affects the variations in the rotor thrust. This has an impact on $M_{Y-\mathrm{BEM}}^{\mathrm{BR}}$, as Fig. 9 (c) shows. The reaction of $\Omega$ to the incoming wind field is a global aerodynamic phenomenon and is largely affected by the rotor wake. The correction model that has the most

influence was found to be the wake memory effect model.

As already mentioned before, we did not include the wake memory effect model in the BEM simulations of this study. Nonetheless, we did some individual calculations with the new wake memory effect model in OpenFAST to see if it was the main driver of the differences between both aerodynamic models.

The result can be seen in Figs. 10 (a) and (b). Including the wake memory effect model (termed DBEMT in the figure)

significantly reduces the oscillation amplitude of $\Omega$ and $\theta$. As a consequence, PSD($M_Y^{\mathrm{BR}}$) for the DBEMT simulations has less



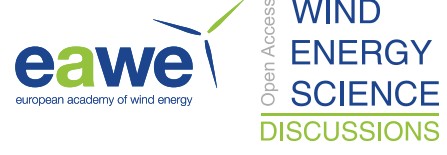

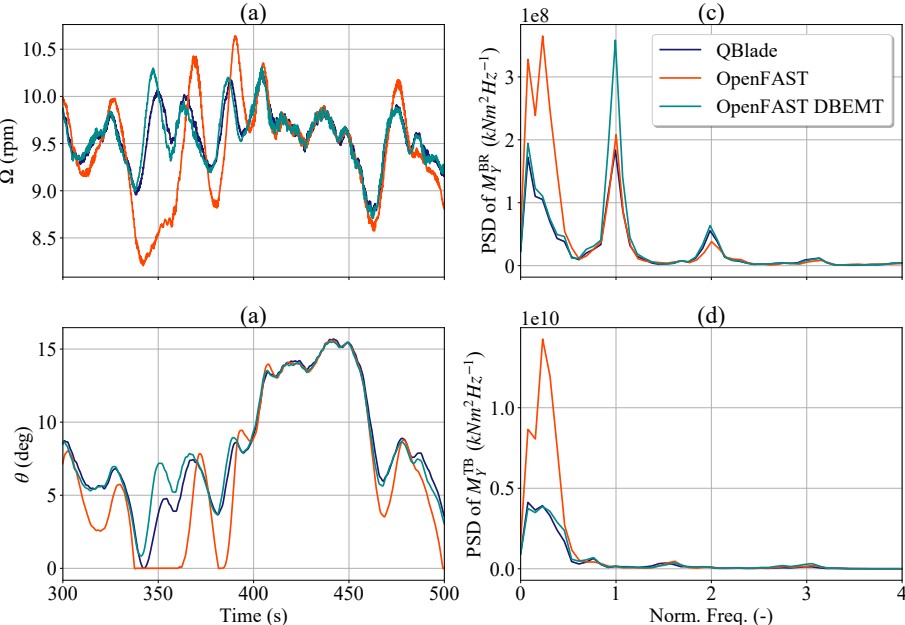

**Figure 10.** Influence of the wake memory effect model on controller behavior and loads for a 14 m/s mean wind speed simulation. (a) Selected $\Omega$ time series; (b) Selected $\theta$ time series; (c) PSD of $M_Y^{\mathrm{BR}}$; (d) PSD of $M_Y^{\mathrm{TB}}$

energy in the low frequency region and is comparable to the PSD of the LLFVW simulation (Fig. 10 (c) ). Surprisingly, turning on the wake memory effect in the DBEMT simulations also increases the 1P contribution of $M_Y^{\mathrm{BR}}$. It is believed that this peak originates from an additional averaging procedure of the axial induced velocity that takes place in the DBEMT simulation. Further analysis is needed to corroborate this assumption.

Returning to Fig. 9, we can see in the subfigures (e) and (f) that for wind speeds in Region C the difference in PSD($M_Y^{\mathrm{BR}}$) between both aerodynamic codes becomes negligible. The principal contribution to the loads in this region is again at the 1P frequency of the PSD, coming mainly from the wind shear. The variability of the controller signals in this region is comparable (Fig. 6) and the contribution of the controller to the PSD($M_Y^{\mathrm{BR}}$) at low frequencies is small. The small difference in the PSD between the BEM and LLFVW calculations is in line with the small relative difference of the normalized $\overline{\mathrm{DEL}}_{\mathrm{1Hz}}(M_Y^{\mathrm{BR}})_{\mathrm{LLFVW}}$

in Region C (Fig. 8 (a) ). The fact that the influence of the different aerodynamic models on $M_Y^{\mathrm{BR}}$ diminishes for higher wind speeds has also been reported in (Jeong et al., 2014).

### 5.3.2   Tower Base Fore-Aft Bending Moment

Of all the considered load sensors, $M_Y^{\mathrm{TB}}$ shows the largest differences in lifetime and 1 Hz DELs. Figure 11 shows the PSD plots for the $M_Y^{\mathrm{TB}}$ sensors in Regions A, B and C. The rows and columns are organized in the same way as in Fig. 9.

For turbulent wind speed calculations in Region A (Fig. 11 (a) ), we can see that the main differences in PSD($M_Y^{\mathrm{TB}}$) from both aerodynamic codes lie close to the 3P frequency. The source of this difference comes mostly from the wind shear, as can



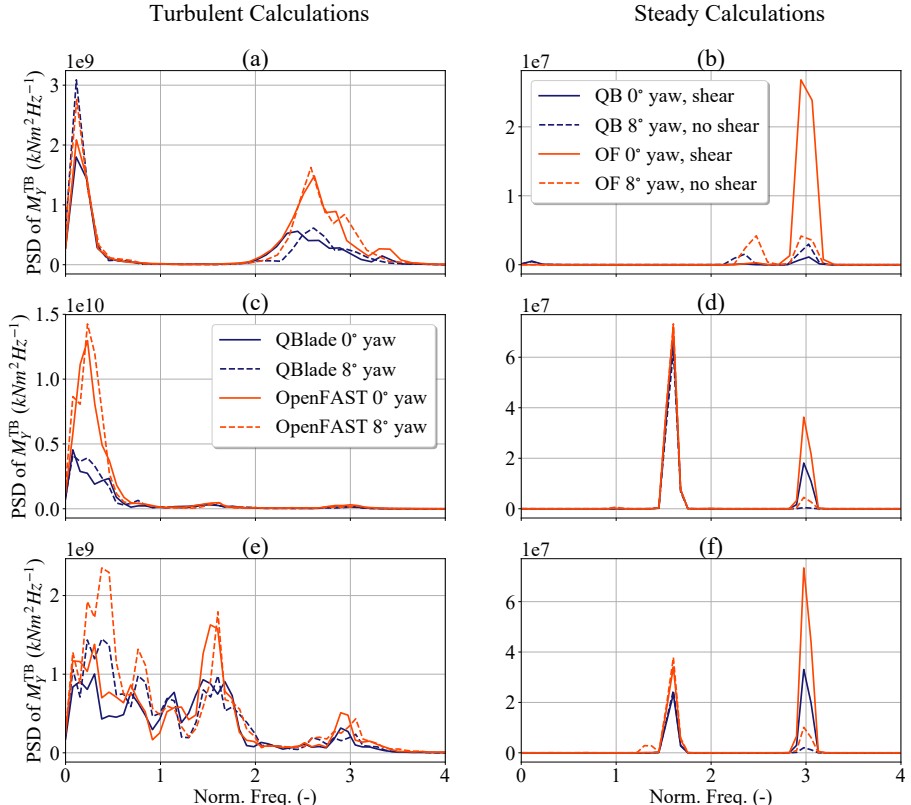

**Figure 11.** Power Spectral Density plots for $M_Y^{\text{TB}}$ at different wind speeds. (a) Turbulent calculations at 8 m/s wind speed; (b) Steady calculations at 8 m/s wind speed; (c) Turbulent calculations at 14 m/s wind speed; (d) Steady calculations at 14 m/s wind speed; (e) Turbulent calculations at 20 m/s wind speed; (f) Steady calculations at 20 m/s wind speed

be seen in Figure 11 (b). The reason for this is as follows. Since the amplitude of the 1P frequency component of $M_{Y-\text{LLFVW}}^{\text{BR}}$ in sheared flow is lower than for $M_{Y-\text{BEM}}^{\text{BR}}$ (Fig. 9 (b) ), the amplitude of the PSD at the tower passing frequency – i.e. 3P – will also be lower for the LLFVW simulations. The fact that the differences in Fig. 11 (a) do not lie exactly on the 3P frequency comes from the varying rotor speed in the simulations. The normalization of the frequencies was done using the average rotor speed of each simulation.

If we now concentrate on Region B simulations, we can see in Fig. 11 (c) that the dominant frequencies in PSD($M_Y^{\text{TB}}$) for all simulations are the low, sub-1P frequencies. It is also this frequency range of the PSD that contains the largest differences between both codes. While there are some differences in the PSD at the 3P frequencies due to wind shear (Fig. 11 (d) ), the contribution of this frequency is several orders of magnitude smaller than the contribution of the low frequency range. As in the case of $M_Y^{\text{BR}}$, the reason for this loading difference can ultimately be linked to the missing memory wake effect in the BEM calculations. If we include the wake memory effect, the differences in the low frequency range of PSD($M_Y^{\text{TB}}$) become negligible, as Fig. 10 (d) shows.




For simulations in Region C the PSD($M_Y^{\text{TB}}$) of both codes is more comparable and at the same time more complicated (Fig. 11 (e)). There are several frequency regions in which the PSD of the BEM simulations is higher than the PSD of the LLFVW simulations. For the 3P frequency, the difference is due to the wind shear (Fig. 11 (f) ) but its contribution to the PSD is small compared to the lower frequencies. The source of the higher amplitudes of the PSD load peaks at low frequencies comes

from the controller action, which is comparable in both codes in this region (Fig. 6) but still somewhat higher for the BEM simulations. While the low frequency peaks are still larger for the BEM simulations, their amplitude stays within a comparable range to the LLFVW peaks. The actual magnitude of the peaks varies depending on the individual simulations. As with $M_Y^{\text{BR}}$, the high convection speed of the wake reduces the axial induced velocity on the rotor disk, decreasing the effect of the different aerodynamic models on the tower base loads. This explains why in Region C the $\overline{\text{DEL}}_{1\text{Hz}}(M_Y^{\text{TB}})$ of the simulations in both

codes are also comparable (Fig. 8 (c) ).

We note that there is a peak in PSD($M_Y^{\text{TB}}$) at about 1.5P frequency in Figs. 11 (d) , 11 (e) and 11 (f). This corresponds to an absolute frequency of 0.25 Hz, which is the natural frequency of the 1st tower fore-aft and side-side mode of the turbine (Bak et al., 2013). In the simulations we saw that the mode was lowly damped in the side-side direction and contributed to the oscillations of $M_X^{\text{TB}}$. The contribution of this mode to PSD($M_Y^{\text{TB}}$) is comparable for both codes, yet the peak for the BEM

simulations is consistently higher. This indicates that the aerodynamic damping of the 1st tower fore-aft mode is higher for the LLFVW simulations than for the BEM simulations.

### 5.3.3 Yaw Bearing Roll Moment

In absolute terms, $M_X^{\text{YB}}$ is the load sensor with the smallest variation in amplitude. So small differences in loading will have a large influence on the relative contribution to the fatigue loads of this sensor. This load component is affected by the generator

torque and by the side-side force acting on the rotor hub. The latter force causes a roll moment due to the vertical offset of the rotor hub to the yaw bearing. A similar analysis was performed for this sensor as it was done for $M_Y^{\text{BR}}$ and $M_Y^{\text{TB}}$, although for brevity only the results will be stated here.

For turbulent simulations in Region A, the main difference in PSD($M_X^{\text{YB}}$) lies in the low frequency range where the controller is active. It is therefore the variability of the generator torque that is the source of the load differences in this region. It could be

argued that the variability of $\Omega$ for this particular wind speed bin is larger in the LLFVW simulations (see Fig. 6 (a) ). Yet the higher variability of the electrical power in Region A for the BEM simulations seen in Fig. 5 (b) indicates that in this region there is a higher fluctuation in the generator torque which causes the higher fatigue loads of $M_X^{\text{YB}}$. The ultimate reason for this difference can again be traced back to the wake memory effect that was not included in the BEM simulations. It is also this phenomenon that is the source of the differences in Region B.

When we consider Region C, we can see in Fig. 8 (b) that the normalized $\overline{\text{DEL}}_{1\text{Hz}}(M_X^{\text{YB}})_{\text{LLFVW}}$ are larger than 1, indicating that the fatigue loads derived from the LLFVW simulations are higher than the ones derived from the BEM simulations. The reason for this is the lowly damped oscillations of the 1st tower side-side mode mentioned above. This side-side oscillation of the tower top is not directly influenced by the aerodynamics. While the relative contribution of the 1st tower fore-aft mode to PSD($M_Y^{\text{TB}}$) is moderate (Fig. 11 (f) ), the relative contribution of the 1st tower side-side mode to PSD($M_X^{\text{YB}}$) is much higher.

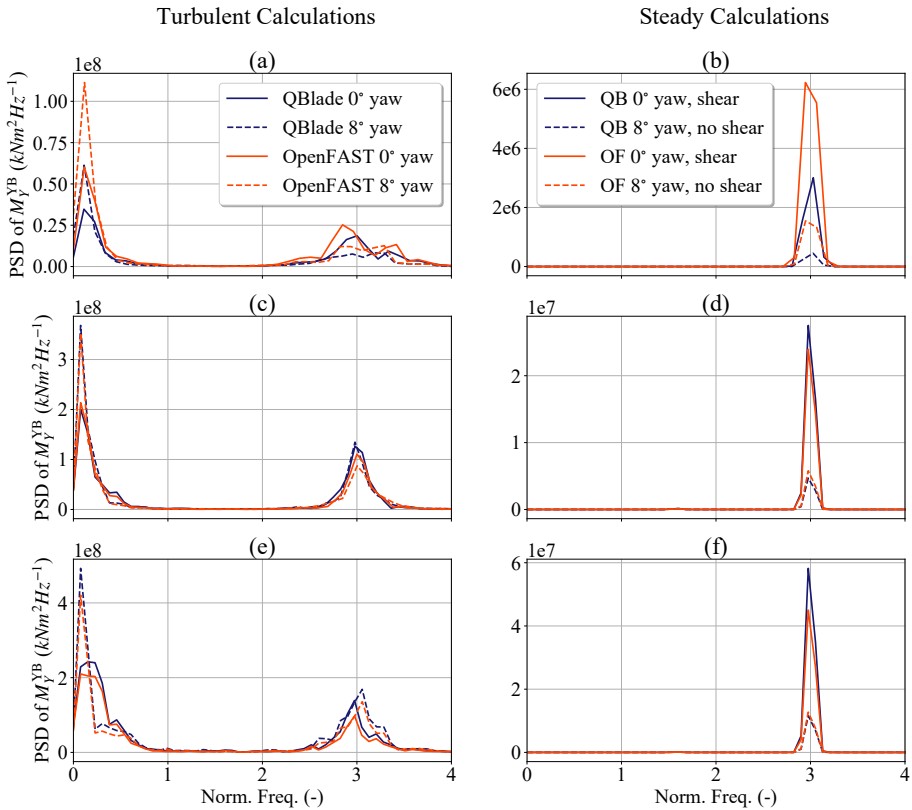

**Figure 12.** Power Spectral Density plots for $M_Y^{\text{YB}}$ at different wind speeds. (a) Turbulent calculations at 8 m/s wind speed; (b) Steady calculations at 8 m/s wind speed; (c) Turbulent calculations at 14 m/s wind speed; (d) Steady calculations at 14 m/s wind speed; (e) Turbulent calculations at 20 m/s wind speed; (f) Steady calculations at 20 m/s wind speed

Because of the small absolute variations of this load signal, the side-side forces present in the hub contribute significantly to the fatigue loads. In our study, BEM simulations show higher oscillations for certain wind speeds and turbulent seeds while in other cases the LLFVW show higher oscillations. Globally, the contributions of the tower side-side deflections even out, as the lifetime DEL of $M_X^{\text{TB}}$ in Fig. 7 shows. For higher wind speeds in particular, the side-side oscillations of the tower top tend to
5  have a higher amplitude in the LLFWV simulations, explaining the higher 1Hz DELs of $M_X^{\text{YB}}$ for the latter aerodynamic code seen in this region.

### 5.3.4 Yaw Bearing Tilt Moment

The last sensor analyzed in this section is $M_Y^{\text{YB}}$. Figure 12 show the PSD plots of $M_Y^{\text{YB}}$ for the same simulations as Figs. 9 and 11.
10    When we consider the results of turbulent simulations in Region A (Fig. 12 (a) ), we can see that there are two clear peaks where $\text{PSD}(M_Y^{\text{YB}})_{\text{BEM}}$ is higher than $\text{PSD}(M_Y^{\text{YB}})_{\text{LLFVW}}$. One is at the 3P frequency and the other at the below-1P frequencies.





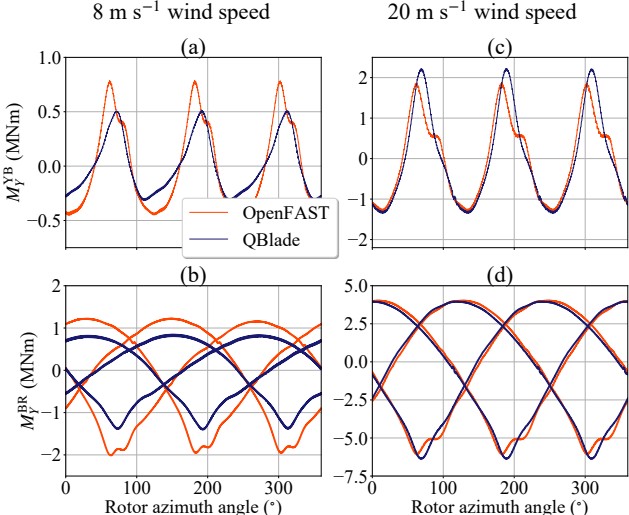

**Figure 13.** Variation of $M_Y^{\mathrm{YB}}$ and $M_Y^{\mathrm{BR}}$ of all three blades as a function of the rotor azimuth angle for steady wind simulations with $0°$ yaw error and wind shear. All signals have a 0 kNm average for better comparison. (a) $M_Y^{\mathrm{YB}}$ for 8 m/s hub wind speed; (b) $M_Y^{\mathrm{BR}}$ for 8 m/s hub wind speed; (c) $M_Y^{\mathrm{YB}}$ for 20 m/s hub wind speed; (d) $M_Y^{\mathrm{BR}}$ for 20 m/s hub wind speed

The latter region corresponding to the frequencies of the time varying turbulent wind and the resulting controller reaction. These peaks can be explained by the fact that the source of $M_Y^{\mathrm{YB}}$ – measured in a non-rotating frame of reference – is the non-uniform distribution of $M_Y^{\mathrm{BR}}$ from the three blades, which is measured in a rotating frame of reference (Burton et al., 2011). In particular, amplitude changes at the 1P frequency of PSD($M_Y^{\mathrm{BR}}$) contribute to amplitude changes at the 0P frequency (or very low frequencies in case of varying wind speed) of PSD($M_Y^{\mathrm{YB}}$). Changes at the 1P frequency of PSD($M_Y^{\mathrm{BR}}$) also contribute to amplitude changes at the 2P frequency of PSD($M_Y^{\mathrm{YB}}$), although the contribution of the loads at this frequency to the fatigue loads of $M_Y^{\mathrm{YB}}$ is negligible for three-bladed turbines. Changes at the 2P frequency in the PSD($M_Y^{\mathrm{BR}}$) contribute to changes at the 1P and 3P frequencies in PSD($M_Y^{\mathrm{YB}}$). Again, only the 3P frequency in PSD($M_Y^{\mathrm{YB}}$) has an important load contribution for this sensor in the case of a three-bladed turbine. As we can see in Figures 9 (a) and (b), the 1P and 2P peaks in the PSD of $M_{Y-\mathrm{BEM}}^{\mathrm{BR}}$ have a higher amplitude than the peaks from $M_{Y-\mathrm{LLFVW}}^{\mathrm{BR}}$. The reason for this differences comes form the effect of the wind shear on the local blade aerodynamics. Wind shear is also the main contributor to the differences in the case of $M_Y^{\mathrm{YB}}$ (Fig. 12 (b) ), although in this subfigure the steady state or 0P load contribution is missing due to the calculation algorithm used to obtain the PSD plots.

For Region B, the qualitative behavior of the PSD changes (Fig. 12 (c) ). Here, the peak in the below 1P frequency region from both codes in the turbulent calculations is comparable. It is at the 3P frequency of the PSD that the LLFVW simulations predict a peak with slightly higher amplitude than the BEM simulations. This is mainly coming again from the wind shear – Fig. 12 (d). The reason for this qualitative change in the PSD can be understood if we consider Fig. 13. This figure shows the variation of $M_Y^{\mathrm{YB}}$ and $M_Y^{\mathrm{BR}}$ for all three blades as a function of the rotor azimuth angle. It is taken from a steady wind





simulation with 0° yaw error and with a wind shear exponent of 0.2. The left column shows the loads for a simulations with 8 m/s hub wind speed and the right column shows the loads for a 20 m/s hub wind speed simulation. All the load signals have a mean value of 0 kNm to allow for a better comparison.

For the simulations in Region A (Figs. 13 (a) and (b) ), there is a clear 3P oscillation of $M_Y^{\mathrm{YB}}$ whose peaks are located at the rotor azimuth angles when the blades pass in front of the tower. The $M_{Y-\mathrm{BEM}}^{\mathrm{BR}}$ of each blade have a higher amplitude in its 1P oscillation (compared to $M_{Y-\mathrm{LLFVW}}^{\mathrm{BR}}$), as we can see in Fig. 13 (b). This causes a larger out-of-plane load imbalance on the rotor and the higher 3P peaks of $M_{Y-\mathrm{BEM}}^{\mathrm{YB}}$ seen in Fig. 13 (a). We can also see in Fig. 13 (b) that there is a small oscillation of $M_Y^{\mathrm{BR}}$ at the azimuth angles when each blade passes in front of the tower and is affected by the velocity deficit due to the tower shadow. $M_{Y-\mathrm{BEM}}^{\mathrm{BR}}$ has a more pronounced oscillation than $M_{Y-\mathrm{LLFVW}}^{\mathrm{BR}}$. These different reactions to the effect of the tower shadow can be traced back to the different ways both codes treat of the local axial induced velocities on the blades.

For simulations in Region C the 3P peaks of $M_Y^{\mathrm{YB}}$ have increased in magnitude and in this case, the amplitude seen in the BEM simulations is smaller than the amplitude in the LLFVW simulations (Fig. 13 (c) ). The lower amplitude peak of $M_{Y-\mathrm{BEM}}^{\mathrm{YB}}$ originates from the effect of the tower shadow on $M_{Y-\mathrm{BEM}}^{\mathrm{BR}}$, as Fig. 13 (d) shows. The small oscillation in $M_Y^{\mathrm{BR}}$ – induced by the tower shadow when each blade passes in front of the tower – is less pronounced, has a higher damping and has a slight shift in the LLFVW simulations compared to the BEM simulations. This is enough to increase the load asymmetry on the rotor and affect the 3P load peak of $M_Y^{\mathrm{YB}}$. It is this effect that is causing the higher 3P peak of the LLFVW simulations in the PSD shown in Figure 12 (d). Although the effect in Region B is not as pronounced as in Region C (depicted in Fig. 13).

Returning to Figure 12, we see in the subfigures (e) and (f) the PSD($M_Y^{\mathrm{YB}}$) for simulations in Region C. The higher amplitude of the 3P frequency peak in the LLFVW simulations is coming from the local effect of the tower shadow velocity deficit on $M_Y^{\mathrm{BR}}$, as we saw above. In addition, there is a small difference in the PSD at the low sub-1P frequency peaks. At those frequencies, the LLFVW simulations have a peak with higher amplitude than the BEM simulations. The cause of this difference is not fully understood and remains open for further investigation.

## 6 Ultimate State Analysis of the Design Load Calculation Results

In last section, we discussed the contribution of the periodic oscillations on the turbine loading. This section considers the extreme events that the turbine sensors experienced in the turbulent wind load calculations. The ultimate state analysis was done for all the sensors listed in Table 3. We analyze the deflection and control signals in the first subsection and the load sensors in the second subsection. The last subsection discusses the differences of the extrema and the reasons behind these differences.

The extreme values presented in this subsection are obtained by taking the maximum and minimum occurring values in the time series of all the simulations. In addition, the extreme values of the blade related sensors – i.e. $M_X^{\mathrm{BR}}$, $M_Y^{\mathrm{BR}}$, $D_X^{\mathrm{BT}}$, $D_Y^{\mathrm{BT}}$ and $\theta$ – are obtained from one blade only. The same blade was considered in the analysis of the BEM and the LLFVW simulations. For this study, it is considered that the extreme events-analysis of one blade is representative of all three blades.





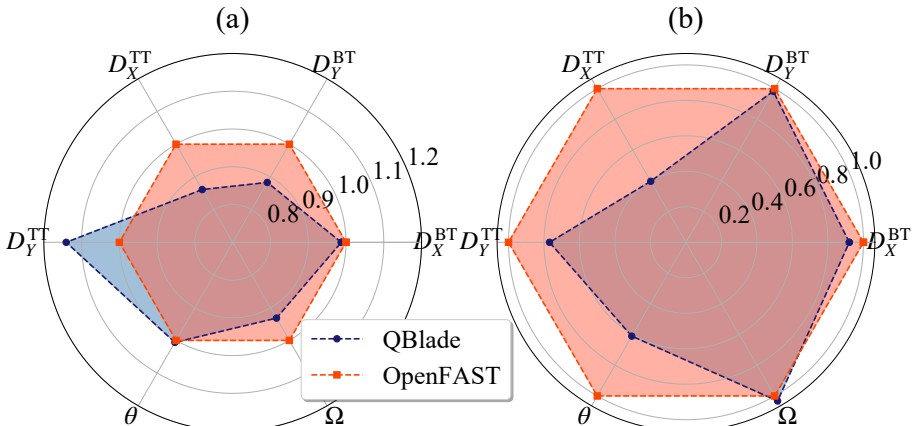

**Figure 14.** Normalized extreme values of deflections and controller signals. (a) Maxima; (b) Minima

Analogously to the fatigue analysis, we will use the notation $\text{Max}(\cdot)_{\text{BEM}}$ / $\text{Min}(\cdot)_{\text{BEM}}$ for the maximum and minimum of a sensor in the BEM simulations. The extrema for the LLFVW simulations will have the corresponding subscript. Although we present the results for all sensors, we will concentrate our discussion and analysis on the out-of-plane related sensors. These sensors are the most directly affected by the differences in the aerodynamic models.

**6.1 Deflections and Controller Signals**

Figure 14 shows the normalized extreme values of the blade tip and tower top deflections as well as the pitch angle and rotor speed. It is clear from this figure that using different aerodynamic models in load calculations also affects the extrema of the considered sensors.

When looking at the blade deflections, it is remarkable to see that the extrema of $D_X^{\text{BT}}$ are very similar in both calculations.
From the higher 1Hz DELs of $M_Y^{\text{BR}}$ in the BEM simulations at wind speeds close to the rated wind speed, we would expect to see blade deflections with higher amplitudes in the BEM simulations and hence larger extrema of $D_X^{\text{BT}}$. While on average the amplitude of $D_X^{\text{BT}}$ in the BEM simulations is larger than in the LLFVW calculations, the normalized value of $\text{Max}(D_X^{\text{BT}})_{\text{LLFVW}}$ is 0.99.

The tower top deflections show larger differences in extreme values from the different calculations than the blade tip de-
flections. If we consider the extrema of the fore-aft deflection, we see that the normalized values of $\text{Max}(D_X^{\text{TT}})_{\text{LLFVW}}$ and $\text{Min}(D_X^{\text{TT}})_{\text{LLFVW}}$ are 0.86 and 0.4, respectively. An indication of the reason behind these large differences can be seen in Fig. 5 (a). The extreme values of the rotor thrust in the BEM calculations are particularly large for the wind speed bin of 16 m/s. Such large values are not present in the LLFVW simulations. These higher thrust forces translate to higher tower deflections and loads.

Finally, Fig. 14 also shows the normalized extreme values of the pitch angle and rotor speed. The maxima of the pitch angle $\theta$ are very similar in both codes. The large relative difference in the normalized $\text{Min}(\theta)_{\text{LLFVW}}$ comes for the fact that


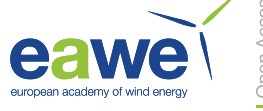


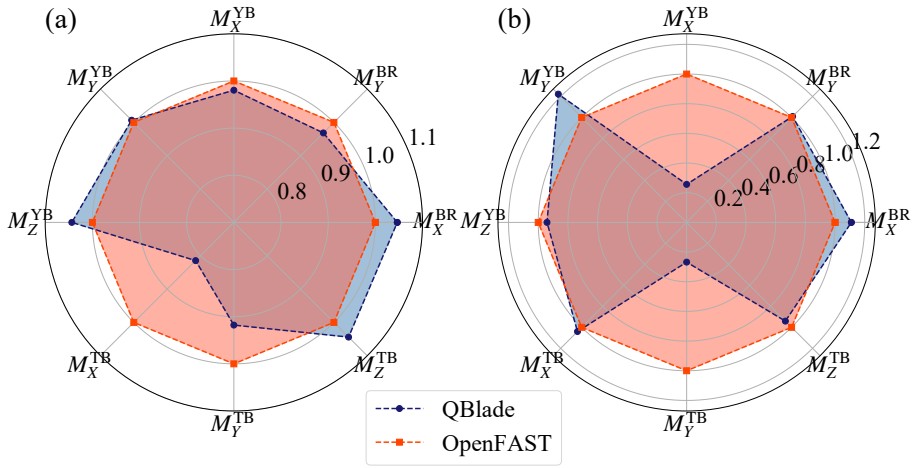

**Figure 15.** Normalized extreme values of turbine load sensors. (a) Maxima; (b) Minima

the minimum $\theta$ is practically $0°$ in both codes. The rotor speed $\Omega$ on the other hand shows a larger difference in the maxima. The normalized $\text{Max}(\Omega)_{\text{LLFVW}}$ is 0.93. An indication for the source of this difference can also be seen in Fig. 5 (d). Here, the maximum of $\Omega$ in the BEM simulations at the 16 m/s wind speed bin is significantly larger than the maxima of $\Omega$ in the LLFVW simulations. An analysis explaining the differences in this section is done in Sect. 6.3.

## 5   6.2   Loads

Performing load calculations with different aerodynamic models also has an impact on practically all the extreme loads of the turbine, as Fig. 15 shows.

Let us start with the blade root loads. We can see in Fig. 15 that the normalized extrema of $M_Y^{\text{BR}}$ are very similar in both calculations. This correlates with the fact that the extreme values of $D_X^{\text{BT}}$ in Fig. 14 were also very similar between both codes.
10   The normalized $\text{Max}(M_Y^{\text{BR}})_{\text{LLFVW}}$ and $\text{Min}(M_Y^{\text{BR}})_{\text{LLFVW}}$ are 0.97 and 1.01, respectively.

In the case of the yaw bearing, the most notable difference in extreme loads occurs for the tilting moment. The normalized $\text{Min}(M_Y^{\text{YB}})_{\text{LLFVW}}$ is 1.22.

For the tower base loads we see that the largest differences in the extrema come from the fore-aft bending moment. The normalized values of $\text{Max}(M_Y^{\text{TB}})_{\text{LLFVW}}$ and $\text{Min}(M_Y^{\text{TB}})_{\text{LLFVW}}$ are 0.92 and 0.26. In the design of tubular axis-symmetric towers,
15   it is usually the resulting extreme bending moment that is one of the design-drivers for the tower. This resulting bending moment at the base is largely affected by $\text{Max}(M_Y^{\text{TB}})$, so a normalized value of 0.92 is quite remarkable. A deeper analysis of these differences in the extreme loads is presented in the next section.



## 6.3 Discussion

As with the fatigue loads, the reason for these differences in the extreme loads must ultimately come from the different aerodynamic models.

In order to limit the extension of this analysis, we will only consider a selection of the sensors. These are: $M_Y^{\text{TB}}$, $D_X^{\text{TT}}$, $M_Y^{\text{BR}}$, $D_X^{\text{BT}}$ and $M_Y^{\text{YB}}$ since they show large deviations and are directly influenced by the aerodynamic loads. The events that cause the extrema of these sensors may also be responsible for the extrema of other sensors. When this is the case, we will include the analysis of the other sensors as well.

While doing the ultimate load analysis, we noted that the extrema of BEM and LLFVW simulations did not necessarily occur in the same simulation or even the same wind speed bin. This can also be seen up to some extend in Fig. 5 where the maxima of rotor thrust and rotor speed for each code occur at different wind bins. In the following analysis we will always present the load case where the highest (absolute) extreme value of the sensors occurred, whether it happened for the BEM calculations or the LLFVW calculations. So for example if the maximum of $M_Y^{\text{TB}}$ was higher for the BEM code, we will include the time series analysis of the BEM load case and show the corresponding LLFVW load case as a comparison. The load case where the maximum of $M_Y^{\text{TB}}$ in the LLFVW simulations occurred will not be analyzed.

### 6.3.1 Tower Loads and Deflections

For the extreme values of the tower sensors, both the maxima and minima of $M_{Y-\text{BEM}}^{\text{TB}}$ and $D_{X-\text{BEM}}^{\text{TT}}$ occurred in the same load case. If we recall Fig. 5, there is an extreme event in the BEM simulations at the 16 m/s wind bin. This extreme event is shown in Fig. 16 and is responsible for $\text{Max}(M_Y^{\text{TB}})_{\text{BEM}}$, $\text{Min}(M_Y^{\text{TB}})_{\text{BEM}}$, $\text{Max}(D_X^{\text{TT}})_{\text{BEM}}$, $\text{Min}(D_X^{\text{TT}})_{\text{BEM}}$ as well as $\text{Max}(\Omega)_{\text{BEM}}$.

As we can see in Fig. 16 (a) there is a sudden increase of the hub wind speed from 10 m/s to about 16 m/s at around 635 s of the simulation time. Several seconds before this sudden gust, $\Omega_{\text{BEM}}$ has dropped to a value below 9 rpm, while $\Omega_{\text{LLFVW}}$ remains in a range between 9 and 10 rpm (Fig. 16 (b) ). The relatively low value of $\Omega_{\text{BEM}}$ for simulation times around 600 s prompts the pitch controller to decrease $\theta_{\text{BEM}}$ to 0° while in the LLFVW simulation, $\Omega_{\text{LLFVW}}$ remains close to $\Omega_R$ and $\theta_{\text{LLFVW}}$ stays around 5° (Fig. 16 (c) ). So when the wind gust arrives, the thrust seen by the turbine rotor in the BEM simulations is much higher, which leads to the maxima of $M_{Y-\text{BEM}}^{\text{TB}}$ and $D_{X-\text{BEM}}^{\text{TT}}$, seen in figures 16 (d) and (e). This is not the case for the LLFVW simulations, mainly because the pitched blades generate less aerodynamic thrust. Moreover, the low values of $\theta_{\text{BEM}}$ also cause the blades to generate more aerodynamic torque when the gust arrives, increasing $\Omega_{\text{BEM}}$ to its maximum value at around 640 s of simulation time (Fig. 16 (b) ). Accordingly, $\theta_{\text{BEM}}$ increases sharply to limit the overshoot of the rotor speed. This in turn decreases the rotor thrust, causing a large amplitude in the return deflection of the tower top (Fig. 16 (e) ). This return deflection is the cause of $\text{Min}(M_Y^{\text{TB}})_{\text{BEM}}$, and $\text{Min}(M_X^{\text{TT}})_{\text{BEM}}$.

The difference in the controller behavior causing these extrema in the BEM simulations can be traced back to the missing wake memory effect in the BEM simulations. Fig. 16 also includes the simulation of this particular load case with the BEM code including the wake memory effect (termed DBEMT in the figure). We can see that by including the wake memory, the

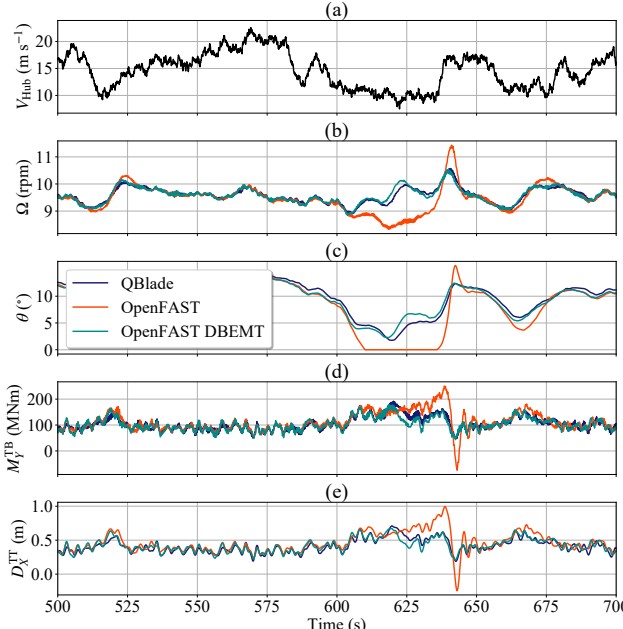

**Figure 16.** Time series of extreme tower event. (a) Wind speed at hub height; (b) Rotor speed; (c) Pitch angle; (d) Tower base fore-aft bending moment; (e) Tower top fore-aft deflection

rotor speed in the BEM simulation stays at values around $\Omega_R$. This in turn leaves the pitch angle of the rotor blades at angles close to $5°$, avoiding the high thrust and torque when the wind gust arrives.

### 6.3.2 Out-of-Plane Root Bending Moment and Tip Deflection of the Blade

A similar analysis as in the previous section was also carried out for $M_Y^{BR}$ and $D_X^{BT}$. For brevity, only the findings will be
5   presented here.

For the BEM simulations, $\text{Max}(M_Y^{BR})_{BEM}$ and $\text{Max}(D_X^{BR})_{BEM}$ occurred for a simulation at the wind speed bin of 12 m/s (although at different times). Similar to Fig. 16, the differences in the blade root loading and tip deflection come from a lower $\theta_{BEM}$ at the moment the wind turbine encountered a small wind gust. The reason for this different controller behavior can ultimately be traced back to the lack of wake memory effect in the BEM simulations.

### 10   6.3.3 Yaw Bearing Tilt Moment

While the normalized $\text{Max}(M_Y^{YB})_{LLFVW}$ in Fig. 15 is very close to 1, the normalized value of $\text{Min}(M_Y^{YB})_{LLVWV}$ reaches a value of 1.22. In order to find the source of this difference, we can refer to Fig. 17. This figure shows a time selection of the load case where both $\text{Min}(M_Y^{YB})_{LLFVW}$ and $\text{Min}(M_Y^{YB})_{BEM}$ occurred. It is a simulation in the 14 m/s wind bin, where large differences in the controller signals are present (Fig. 6). The time instant where $\text{Min}(M_Y^{YB})_{LLFVW}$ occurs is 237.1 s (Fig. 17 (e) ). The time
15   instant of $\text{Min}(M_Y^{YB})_{BEM}$ is almost at the same moment: it happens at a simulation time of 237.6 s. Around these time instants,





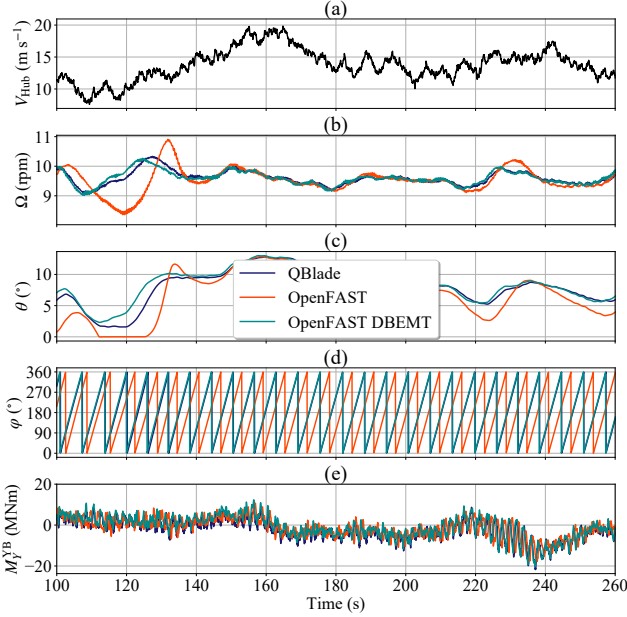

**Figure 17.** Time series for $\text{Min}(M_Y^{\text{YB}})_{\text{LLFVW}}$ event. (a) Wind speed at hub height; (b) Rotor speed; (c) Pitch angle; (d) Rotor azimuth angle; (e) Yaw bearing tilt moment

there is no particular event in the hub wind speed signal and – except for a larger oscillation of the $\Omega_{\text{BEM}}$ and $\theta_{\text{BEM}}$ around 230 s – the controller signals have comparable values (subfigures (a) - (c) ). It is the difference in the rotor azimuth angle $\varphi$ that is causing the larger amplitude of $\text{Min}(M_Y^{\text{YB}})_{\text{LLFVW}}$ at 237.1 s (subfigure (d) ). If we consider the time range between 100 s and 120 s, the $\varphi_{\text{BEM}}$ and $\varphi_{\text{LLFVW}}$ are fairly close together with temporal phase shift of about 1.3 s. Around the same time range,

5   a large oscillation of $\Omega_{\text{BEM}}$ starts. The result of the oscillation is a major shift of the azimuth angles between both codes. The temporal phase shift $\varphi_{\text{BEM}}$ and $\varphi_{\text{LLFVW}}$ after this oscillation is 2.5 s, or the equivalent of $144°$. This phase shift is kept more or less constant until the time of the extreme event. Because of the different rotor azimuth position, the turbulent wind distribution that the blades in the LLFVW simulation see is not the same as in the BEM simulations. In this particular case, it happened to be a distribution that caused a sharper negative value $M_Y^{\text{YB}}$ in the LLFVW simulation.

10    The ultimate reason for the larger oscillations in $\Omega_{\text{BEM}}$ – which cause the large shift in the azimuth angles – is again the lack of the wake memory effect in the BEM simulation. Figure 17 also includes the BEM simulation with the wake memory effect. We can see that if we include this correction, $\Omega_{\text{BEM}}$ follows much closer $\Omega_{\text{LLFWV}}$. Hence, the temporal phase shift of $\varphi_{\text{BEM}}$ and $\varphi_{\text{LLFVW}}$ is very small leading to closer values of $\text{Min}(M_Y^{\text{YB}})_{\text{LLFVW}}$ and $\text{Min}(M_Y^{\text{YB}})_{\text{BEM}}$.



## 7   Conclusions

In this paper we analyzed the effect of two different aerodynamic models on the performance and especially on the loads of the DTU 10 MW RWT. The first aerodynamic model – implemented in the aeroelastic simulation software OpenFAST – is the BEM model, the standard model used in the industry. The second aerodynamic model – implemented in TU Berlin's aeroelastic
software QBlade – is the LLFVW model.

We did a baseline comparison of both codes by calculating the performance of the turbine under constant uniform wind speeds, where the differences between both aerodynamic models are the smallest. The performance coefficients of the turbine simulated with both codes were similar for all relevant wind speeds where the turbine is in power production. The largest differences were seen at wind speeds below rated wind speed, where the axial induction factor plays an important role. Including
wake coarsening measures to speed up the LLFVW simulations as well as elasticity did not have a significant impact on the performance of the wind turbine.

We also simulated the wind turbine under turbulent wind conditions following the requirements of the IEC 61400-1 ed.3 DLC groups 1.1 and 1.2. The average performance of the turbine in the turbulent wind simulations is comparable to the performance in the idealized simulations with constant uniform wind speed. Yet there is considerable variation in the thrust
and power of the turbine due to the unsteady aerodynamic phenomena present in the turbulent wind load calculations. Those variations are more marked in the BEM simulations than in the LLFVW simulations, with the former showing a higher activity in the controller signals – i.e. the rotor speed and the pitch angle. This leads to considerable differences in the fatigue and extreme loads of the turbine.

In order to quantify the differences in the fatigue loads, we carried out a fatigue analysis that includes the lifetime DELs and
the per wind bin-averaged 1 Hz DELs of selected load sensors of the turbine. For the lifetime DELs, the LLFVW simulations show a 4% decrease in $\mathrm{DEL_{Life}}(M_Y^{\mathrm{BR}})$ and a 14% decrease of $\mathrm{DEL_{Life}}(M_Y^{\mathrm{TB}})$ compared to the BEM simulations. Analyzing the averaged 1Hz DELs, we found that the wind speed bins between 6 and 16 m/s contribute the most to the decrease in the sensors' fatigue loads in the LLFVW simulations. For bins with higher wind speeds, the differences in fatigue loads of $M_Y^{\mathrm{BR}}$ and $M_Y^{\mathrm{TB}}$ between both codes diminish. Further analysis showed that the main contributors to the differences in the fatigue
loads of the sensors are the different way the sheared inflow affects the local blade aerodynamics in each code and the missing wake memory effect model in the BEM calculations. The latter contributed to higher variations in $\Omega_{\mathrm{BEM}}$ and $\theta_{\mathrm{BEM}}$ – specially at wind speeds around rated wind speed – that influenced the out-of-plane loading of the turbine.

For the yaw bearing moment, we found that the LLFVW simulations predicted an increase of 4% and 2% in $\mathrm{DEL_{Life}}(M_Y^{\mathrm{YB}})$ and $\mathrm{DEL_{Life}}(M_Z^{\mathrm{YB}})$, respectively. Analyzing the contributions of individual wind speed bins on the 1Hz DELs of $M_Y^{\mathrm{YB}}$ revealed
that for wind speeds up to 12 m/s, the LLFVW simulations predict a decrease in 1 Hz DELs. For wind speed bins of 14 m/s and higher the trend inverses and the LLFVW simulations predict higher 1Hz DELs than BEM simulations. Looking for the reason of this behavior, we found that for bins with low wind speeds, the difference in the way the sheared inflow affects the local blade aerodynamics in both codes was the main contributor of the higher fatigue loads from the BEM simulations for $M_Y^{\mathrm{YB}}$. When we consider wind speed bins with higher wind speeds, we found that one of the reasons of the higher fatigue loads in





the LLFVW simulations is the influence of the velocity deficit due to the tower shadow on the local blade aerodynamics. The local deficit causes a larger load asymmetry on the rotor in the LLFVW simulations when the blades pass in front of the tower, leading to higher amplitude oscillations in the yaw bearing tilt moment. While it is one contributing phenomenon, further research is needed to completely understand what is causing this trend of higher $M_Y^{\mathrm{YB}}$ fatigue loads at high wind speeds.

5    We also did an ultimate state analysis on the results of the turbulent wind load calculations. For the out-of-plane loads and deflections of the tower and blade, we found that the BEM simulations predicted higher extrema than the LLFVW simulations. The maxima of $D_{X-\mathrm{BEM}}^{\mathrm{TT}}$ and $M_{Y-\mathrm{BEM}}^{\mathrm{TB}}$ are 14% and 8% higher than their respective maxima in the LLFVW simulations. As for the blade sensors, we found that the maximum $D_{X-\mathrm{BEM}}^{\mathrm{BT}}$ and $M_{Y-\mathrm{BEM}}^{\mathrm{BR}}$ are 1% and 3% higher than their respective maxima in the LLFVW simulations. The reason for these differences could be traced back to the missing wake memory effect in the BEM simulations, which caused large differences in the behavior of turbine controller and hence the loading. The missing wake memory effect in the BEM simulations was also the reason for the differences in $\mathrm{Min}(M_Y^{\mathrm{YB}})$ between both codes. In the case of this sensor, the different aerodynamic models also affected the controller behavior increasing the minimum of the LLFVW simulations by 22%.

The results of this paper show that there are significant differences in the fatigue and extreme loads if we use a higher order aerodynamic model in the load calculations. In order to improve our quantification of the load differences, future work will include the wake memory effect in the BEM calculation. This correction model was one of the major contributors to the loading differences between both codes. Including it in future evaluations will ensure a fairer comparison between both models.

Future work will also include simulations with a higher-order representation of the structural dynamics. By including the torsional degree of freedom, we will be able to model the flap-twist coupling that greatly influences the loads on the turbine. In order to better quantify the differences in extreme loads, more DLC groups from the current guidelines and standards should be included. Performing an ultimate state analysis of the IEC 64100-1 DLC 1.1 and 1.2 groups gave us some insight into the influence of the aerodynamic codes on the extreme loads. Including DLC groups that are known to induce design driving extreme loads on the turbine will help us understand and quantify better of the effect of higher-order aerodynamic models on the extreme loads.

25    *Code and data availability.*  Both OpenFAST and QBlade are open source codes available online. The latest version of OpenFAST is available at https://github.com/OpenFAST. The latest version of QBlade is available at https://sourceforge.net/projects/qblade/. The version of QBlade used in this paper that includes the structural model will be made available soon. The time series for the BEM and LLFVW calculations used in this paper are stored in the OpenFAST binary format. They can be made available upon request.

## Appendix A:  Wake Coarsening Parameters for the LLFVW Simulations

30   This appendix contains the wake coarsening parameters we used in our LLFVW simulations. They are summarized in Table A1.





**Table A1.** Wake coarsening parameters for aerodynamic and aeroelastic LLFVW simulations

| Simulation type | Wind speed range | Near-wake | Mid-wake | Far-wake | Wake cut-off | Mid-wake factor | Far-wake factor |
|---|---|---|---|---|---|---|---|
| Aerodynamic | 4-25 m/s | 10 revs | 10 revs | 1 rev | 21 | 2 | 3 |
| Aeroelastic | 4 m/s | 0.5 revs | 5.5 revs | 12 revs | 18 revs | 3 | 3 |
| | 6 m/s | 3 revs | 5 revs | 10 revs | 18 revs | 2 | 2 |
| | 8 - 10 m/s | 1 rev | 2 revs | 7.7 revs | 10.7 revs | 3 | 4 |
| | 12 m/s | 0.5 revs | 0 revs | 8 revs | 8.5 revs | 2 | 2 |
| | 14 - 20 m/s | 0.5 revs | 0 revs | 7.5 revs | 8 revs | 2 | 2 |
| | 20 - 24 m/s | 0.5 revs | 0 revs | 6.5 revs | 7 revs | 2 | 2 |

*Author contributions.* S. Perez-Becker prepared the manuscript with the help of all co-authors. D. Marten is the main developer of QBlade and developed the aerodynamic code. J. Saverin developed the structural code. S. Perez-Becker and F. Papi performed the calculations and the analysis of the results. A. Bianchini and C.O. Paschereit provided assistance with the paper review.

*Competing interests.* The authors declare that they have no conflict of interest.

5 *Acknowledgements.* S. Perez-Becker wishes to thank WINDnovation Engineering Solutions GmbH for supporting his research.



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
