# Peer review of "Is the Blade Element Momentum Theory overestimating Wind Turbine Loads? – A Comparison with a Lifting Line Free Vortex Wake Method"

_Wind Energy Science, 2019_

## Referee Comment (RC1) · Georg Raimund Pirrung (Referee) · 4 Dec 2019

Dear authors,

I think this is a very good and important article! It is great that things are really investigated in detail to explain the phenomena that you observe. Also the idealized aerodynamics only comparison is a good idea to access basic model behavior before going ahead with the turbulent simulations. In my opinion the article could be a bit more nuanced about what a BEM model is. For example the title seems a bit exaggerating

to me - in principle you are only investigating if a particular BEM code is overpredicting loads, not the BEM code family as a whole. Also, as you state in the article, it is not a really fair comparison for the loads if the dynamic inflow model is deactivated.

Very recently we did a study of a grid based BEM model against an azimuthally averaged BEM model, both with dynamic inflow (see 'Implementation of the Blade Element Momentum Model on a Polar Grid and its Aeroelastic Load Impact' on wind energy science discussions, currently in proofreading). The outcome of that work is that the difference between a polar grid and an azimuthally averaged BEM model on DLC1.2 blade root flapwise fatigue loads is in the order of 8-10%. This is a similar order of magnitude as the difference between lifting line code and the BEM code you show in your article. The main reasons for these differences are a local dynamic inflow (vs an annular averaged) and better reaction of the induced velocity to shear and sampled turbulence, which is also very well in line with your conclusions.

Based on this difference we observed between different BEM implementations it might be too general to state that 'BEM overestimates loads'. Maybe you could mention our findings in your article or state that different BEM implementations car give different results and 'the BEM model' doesn't exist.

But again, I really like the very thorough approach and the detailed descriptions and analysis of the observed phenomena and I think your article is a very good contribution to the field of wind turbine aerodynamics.

Please find more detailed comments below.

Best regards,

Georg

- page 2 line 3: change the sentence to 'In the case of turbulent wind simulations, several repetitions of individual DLCs with different turbulent wind realizations are required...'. Without that it seems strange that simulations have to be repeated.

- page 3 line 33: The near wake model is not really a lifting line free vortex wake model. It is a simplified lifting line model where the vortices follow helical paths. The helix angles can vary in time, but the vorticity is definitely not as free to move around as in a true free wake model.

- page 4 line 1: than instead of that

- page 5 line 16: This explanation of a dynamic stall model is maybe a bit misleading, because it seems like it is applied on the static airfoil data in a preprocessing step. Can you rephrase this to explain better what the model does ( for example applying time filters on the dynamic angle of attack and the separation point on the airfoil).

- page 8 line 2: You mention that tilt angle violates assumptions in BEM theory. But in practice tilt angle is no different from a yaw so as such it should be handled by the yaw correction model. Also, cone and radial induction can be taken into account in a BEM model, see our recent paper.

- Figure 3: Maybe I missed it but did you use fully turbulent or transition polars?

- page 12: You mention that the pitch angles agree in all computations, and they don't change when going from aerodynamic to aeroelastic simulations. It might be worth mentioning that that is probably only true because blade torsion is not included.

- page 16, line 7: Do you have an idea why the generator speed drops below the minimum generator speed in the BEM simulations? I think that the controller should prevent that.

- page 19: I am not quite sure if it is correct to say that the influence of different aerodynamic models is decreasing with higher wind speed. At high wind speeds, the load distribution changes when going to a higher fidelity model because the

strongest vortices are no longer necessarily trailed from the tip. Thus the common tip loss corrections for BEM models are not doing their job anymore and BEM load distributions differ more from high fidelity results than at low wind speeds. See the attached Figure from 'A coupled near and far wake model for wind turbine aerodynamics' (Wind Energy). Moreover, we found a large difference in fatigue loads for polar grid BEM vs annular average BEM at high wind speed in 'Implementation of the Blade Element Momentum Model on a Polar Grid and its Aeroelastic Load Impact'.

[Figure]

**Figure 12.** Comparison of axial and tangential forces in normal operation at 8 (top) and 25 m s$^{-1}$ (bottom) of a blade element momentum (BEM) model, the coupled near and far wake model and full rotor CFD.

**Fig. 1.**

---

## Referee Comment (RC2) · Emmanuel Branlard (Referee) · 30 Dec 2019

The paper performs a thorough comparison between two lifting line models: the BEM model of AeroDyn, and the free-wake model of QBlade. The paper presents results from steady state and turbulent simulations, looking at damage equivalent loads and extremes. The results and approach are presented in a clear way, the analyses are precise and discussed in details. I congratulate the authors for the work put in this paper, there is potential for a great paper.

[Figure]

I have the following general comments: - Dynamic wake/inflow model: It is true that the Dynamic wake model of OpenFAST is not documented, but I'm afraid this model cannot be discarded for the fatigue analyses. The study would not be fair without it... I'm aware that this will require more time and might entirely change the results and analyses, but I'm afraid that discarding this model makes the study less valuable. The model acts as a filter, which filters high frequencies out and introduces a phase shift in the signal. Without it, the BEM code uses quasi-steady induction factors, with a high and unrealistic frequency content. The DBEMT model of OpenFAST is an implementation of the dynamic wake model of Oye (presented in the report of Snel and Shepers of the book of Martin Hansen). Please consider including the DBEMT model and updating the results of the paper. I'd be happy to assist you if you need further help or documentation. I would recommend using a different time constant for each mean wind speed.

- Aerodynamic differences. As you mention, the elastic and servo parts are the same for both models, the only thing affecting the results are differences of the aerodynamic model. The paper follows a nice scientific approach, yet it seems that there is a gap between the stiff and steady simulations, and the elastic unsteady simulations. It would be valuable to investigate the key aerodynamic differences between the models, using stiff unsteady simulations (e.g. performing a sweep at different yaw angles, studying response to wind steps, or looking at azimuthal variations of inductions similar to fig 13). If possible, these results could be compared to CFD, measurements, or, other BEM implementations. Such results could then be used to interpret the results of the full aero-servo-elastic simulations. This again would require more work, but I think it would be valuable to focus on this first, and maybe present the fatigue analyses in a separate paper... Even if you chose to stay with the current structure, I think it is important if it is stressed that the differences are between two specific implementations, the BEM from OpenFAST and the free vortex wake from QBlade. In light of this, I would think you might want to revise the title to highlight that the results are specific to these two implementations, unfortunately, the title would be less catchy..

- Length of the paper: Despite the careful and valuable analyses and discussions, I believe the paper could be considerably reduced in length. Here are some ideas to reduce the length: The literature review and presentation of the models can be significantly reduced. The results of section 5 are a bit repetitive. You may consider presenting one of the plot for one sensor, and then simply focus on the key conclusions that you drew for the other sensors (maybe summarizing them in a table). You are thorough in your analyses, and I believe the reader will trust your conclusions without having to see the plots. Also, I would think you can remove some text that describe the figures, and move more rapidly to their discussions. The reader might get lost in such a level of details, and I would advise to focus more on the story and key conclusions of the paper.

I enclose some specific comments below. I hope that addressing these general and specific comments will improve the quality of the paper. Clearly a lot of work has been done, and I would be happy to review a revised version of the manuscript. Good luck for the work,

Emmanuel Branlard

p1 l1: "state of the art" might not be appropriate -> "common" maybe (CFD would be state of the art)

p1 l3: BEM does not only simplify the rotor-aero (also wake and inflow). To some extent the rotor aero are the same for BEM and FVW

p1 l19: the "wake memory" effect need to be included for this study, otherwise the results won't be fair

p3: very nice literature review, but quite long, maybe a page could be removed.

p5 l13: "yawed-inflow condition" -> or "yawed and tilted conditions"

p5 l24: the dynamic wake model is the model of {\O}ye

p5: the model description could be condensed to a smaller list with references to shorten the paper.

p6: Can you detail the core model you are using? In particular, please mention how you determine the core size of the bound and wake vorticity? How are these parameters determined as function of the discretization?

p8 l3: "in the rotor plane": It can be argued that the main issue comes from the fact that the annuli are assumed to be independent, not so much that the momentum balance was determined in a plane.

p8 l9: ElastoDyn does not rely solely on Euler-Bernoulli beam theory, it also includes corrections to account for geometric non-linearities.

p10 l1: Feel free to contact me if you want ore information on the DBEMT model. I believe this model needs to be used for the current study.

p10 l14: It would be valuable to present aerodynamic performances of the FVW&BEM at exactly the same operating conditions (RPM/Pitch) instead of using the controller. Presenting radial distribution of axial and tangential inductions along the span at different operating point will reveal the aerodynamic differences between the two models. The inductions are the main variables of lifting-line codes (such as BEM and LLFVW).

p11 tab2: You may consider using "steady" instead of "constant"

p11 l20: "Purely aerodynamic calculations" -> you could replace by "CFD" or something similar maybe?

p13 tab3: The fact that only DLC1.2 is used might indeed make the comparison of the ultimate load difficult, you can consider using a gust case. This would avoid the some of the discussions of section 6 related to the different responses of the two models, where maxima occur at different simulations.

p13 l16: Though I like the extra information on the plot, you can maybe mention that 6

points are likely not statistically significant.

p17 l5: How do your results compare to other studies?

p19 l17: The main effect there is the small amplitudes of the inductions, not the pitch angle. Probably what is meant here is that the angle of attack is mainly composed by the rotational speed and the undisturbed wind, since the inductions are small, and hence the rotor performances are not strongly affected by the aerodynamic model.

p20 l7: Both the BEM implementation from OpenFAST and vortex code have limitations in yawed inflow. For the vortex code, the "wake is going up" in skewed inflow (I discuss this in a paper entitled "Aeroelastic large eddy simulations using vortex methods: unfrozen turbulent and sheared inflow"). On the other hand the limitations of the OpenFAST BEM code are more inherent the implementation choices (coordinate systems used for the axial and tangential inductions, and choice of determination of the wake skew angle), these choices are made differently in different codes. You may want to mention this somewhere in the paper. This is why in my general comments I mention that the observations made are specific to the implementations, and it would be highly valuable to present some "stiff" simulations comparing some key aerodynamic components of the models (against, CFD, measurements, or other BEM codes).

p22 l1-4: How did you determine the time constant for DBEMT here? This might need to be adapted since it does not filter the high frequencies enough.

p23 l12: Is wake memory actually included in Fig 10d? I might have missed it. Or is the figure reference wrong maybe?

p27 l2: "have a mean value of 0" -> "have been adjusted to a mean value of 0", maybe?

p27 l15: DBEMT will also introduce a phase shift, maybe similar to the LLFVW code, or not..

p27 l24: "In last section" -> "In the previous section", or, "In section ..."

p30 l21-29: The comparison of the two simulation might be too difficult (or "anecdotic") since the wind turbine is indeed a highly non-linear system, and both aerodynamic models are behaving quite differently here. I would recommend using a more deterministic case like a gust for the study. Figures 14-15 are still interesting and valuable.

---

## Author Comment (AC1) · 7 Feb 2020

Dear reviewers,

thank you for the detailed and qualified reviews of our paper. Your suggestions did help us to improve it. Please find below the point-to-point reply (in blue-colored text) to your comments.

○○○○○○○○○○○○○○○○○○○○○○○○○○○○○○○○○

**Referee 1 (G. Pirrung)**

Dear authors,

I think this is a very good and important article! It is great that things are really investigated in detail to explain the phenomena that you observe. Also the idealized aerodynamics only comparison is a good idea to access basic model behavior before going ahead with the turbulent simulations. In my opinion the article could be a bit more nuanced about what a BEM model is. For example the title seems a bit exaggerating to me - in principle you are only investigating if a particular BEM code is overpredicting loads, not the BEM code family as a whole.

Thank you for your appreciation of the paper. We do agree that the first draft of the paper could give the impression of comparing the BEM method and the LLFVW method in general, while we actually only compared the one particular implementation of each aerodynamic method in this paper. Based on your right criticism, we changed the manuscript title into "Is the Blade Element Momentum Theory overestimating Wind Turbine Loads? - An Aeroelastic Comparison between OpenFAST's AeroDyn and QBlade's Lifting Line Free Vortex Wake Method". We also changed the introduction, discussion and conclusions so that it is clearly stated that the study only compares two particular codes.

Also, as you state in the article, it is not a really fair comparison for the loads if the dynamic inflow model is deactivated. Very recently we did a study of a grid based BEM model against an azimuthally averaged BEM model, both with dynamic inflow (see 'Implementation of the Blade Element Momentum Model on a Polar Grid and its Aeroelastic Load Impact' on wind energy science discussions, currently in proofreading). The outcome of that work is that the difference between a polar grid and an azimuthally averaged BEM model on DLC1.2 blade root flapwise fatigue loads is in the order of 8-10%. This is a similar order of magnitude as the difference between lifting line code and the BEM code you show in your article. The main reasons for these differences are a local dynamic inflow (vs an annular averaged) and better reaction of the induced velocity to shear and sampled turbulence, which is also very well in line with your conclusions. Based on this difference we observed between different BEM implementations it might be too general to state that 'BEM overestimates loads'. Maybe you could mention our findings in your article or state that different BEM implementations car give different results and 'the BEM model' doesn't exist. But again, I really like the very thorough approach and the detailed descriptions and analysis of the observed phenomena and I think your article is a very good contribution to the field of wind turbine aerodynamics. Please find more detailed comments below.

We also felt that the comparison without the dynamic inflow model in OpenFAST was unfair. Based on the suggestion of referee 2 and with his help, we re-did the comparison using the dynamic inflow model. The new version of the manuscript now has the results and analysis of the OpenFAST calculations with the dynamic inflow model.

Thank you also for pointing out your paper on the BEM method based on a polar grid. Our updated results lie closer to the results from your paper when we include the dynamic inflow in the BEM calculations. We mention the results of your paper in the discussion section, which helped to explain better our results.

• page 2 line 3: change the sentence to 'In the case of turbulent wind simulations, several repetitions of individual DLCs with different turbulent wind realizations are required...'. Without that it seems strange that simulations have to be repeated.

We changed the sentence to match the suggestion.

• page 3 line 33: The near wake model is not really a lifting line free vortex wake model. It is a simplified lifting line model where the vortices follow helical paths. The helix angles can vary in time, but the vorticity is definitely not as free to move around as in a true free wake model.

We changed the description from 'LLFVW model' to 'lifting line vortex method' to make clear that this model is not a lifting line free vortex wake.

• page 4 line 1: than instead of that

Corrected.

• page 5 line 16: This explanation of a dynamic stall model is maybe a bit misleading, because it seems like it is applied on the static airfoil data in a preprocessing step. Can you rephrase this to explain better what the model does (for example applying time filters on the dynamic angle of attack and the separation point on the airfoil).

To keep the theory section brief (suggested by referee 2), we left the most of the explanation of the engineering correction models out, while we cited the AeroDyn manual for the detailed explanation of the models.

• page 8 line 2: You mention that tilt angle violates assumptions in BEM theory. But in practice tilt angle is no different from a yaw so as such it should be handled by the yaw correction model. Also, cone and radial induction can be taken into account in a BEM model, see our recent paper.

We removed the mention of the tilt angle. We included the reference to the Madsen et al. 2020 paper (BEM on polar grid) and we changed the statement that the particular LLFVW implementation is more accurate in load prediction than the particular BEM implementation.

• Figure 3: Maybe I missed it but did you use fully turbulent or transition polars?

The polars used for our calculations come from the DTU10MWReferenceWindTurbine.xls V1.04, which we downloaded from http://dtu-10mw-rwt.vindenergi.dtu.dk/. It references the DTU Wind Energy Report-I-0092. In this report, we could not find a clear indication on which boundary layer turbulence model was finally chosen to generate the polars (section 3.2.1), even though we presume that the fully turbulent one was used.

• page 12: You mention that the pitch angles agree in all computations, and they don't change when going from aerodynamic to aeroelastic simulations. It might be worth mentioning that that is probably only true because blade torsion is not included.

We included a sentence mentioning the absence of blade torsional degree of freedom as a possible source of identical pitch angles.

• page 16, line 7: Do you have an idea why the generator speed drops below the minimum generator speed in the BEM simulations? I think that the controller should prevent that.

The drop of the generator speed below the minimum generator speed occurs in both models for the same simulations (i.e. 4 m/s with the same turbulent wind seed). In this simulation, the wind speed drops below 2 m/s for more than 90 s. The generator torque reaches 0 Nm to keep the minimum generator speed but there is simply not enough aerodynamic torque to keep the generator speed at the minimum speed level. That is why we get a generator speed lower than the controller-minimum.

We do not include this explanation in the paper to keep the discussion brief and focus on the main story.

• page 19: I am not quite sure if it is correct to say that the influence of different aerodynamic models is decreasing with higher wind speed. At high wind speeds, the load distribution changes when going to a higher fidelity model because the strongest vortices are no longer necessarily trailed from the tip. Thus the common tip loss corrections for BEM models are not doing their job anymore and BEM load distributions differ more from high fidelity results than at low wind speeds. See the attached Figure from 'A coupled near and far wake model for wind turbine aerodynamics' (Wind Energy). Moreover, we found a large difference in fatigue loads for polar grid BEM vs annular average BEM at high wind speed in 'Implementation of the Blade Element Momentum Model on a Polar Grid and its Aeroelastic Load Impact'.

We agree with your comment. The aerodynamic load distribution along the blade at high wind speeds changes and this has an influence on the local blade loading. We obtained similar results in reference Saverin et al. 2016b. Our statement focused on the controller behavior and on the considered load sensors. The controller behavior in particular reacts to global changes in the rotor aerodynamics, affected by the effective induced axial velocity of the rotor. Since the induction factors are small in this region, we stated that the influences of the aerodynamic models decreased. We changed the sentence in the paper to reflect this point more clearly.

Based on your suggestion and on the one from reviewer 2, we re-ran the BEM simulations using the dynamic inflow model. The results for the fatigue load comparison on the out-of-plane blade root bending moment now agree much better with the results from the paper 'Implementation of the Blade Element Momentum Model on a Polar Grid and its Aeroelastic Load Impact'.

**Referee 2 (E. Branlard)**

The paper performs a thorough comparison between two lifting line models: the BEM model of AeroDyn, and the free-wake model of QBlade. The paper presents results from steady state and turbulent simulations, looking at damage equivalent loads and extremes. The results and approach are presented in a clear way, the analyses are precise and discussed in details. I congratulate the authors for the work put in this paper, there is potential for a great paper.

We would like to thank sincerely the reviewer for his appreciation of our study.

I have the following general comments:

- Dynamic wake/inflow model: It is true that the Dynamic wake model of OpenFAST is not documented, but I'm afraid this model cannot be discarded for the fatigue analyses. The study would not be fair without it... I'm aware that this will require more time and might entirely change the results and analyses, but I'm afraid that discarding this model makes the study less valuable. The model acts as a filter, which filters high frequencies out and introduces a phase shift in the signal. Without it, the BEM code uses quasi-steady induction factors, with a high and unrealistic frequency content. The DBEMT model of OpenFAST is an implementation of the dynamic wake model of Oye (presented in the report of Snel and Shepers of the book of Martin Hansen). Please consider including the DBEMT model and updating the results of the paper. I'd be happy to assist you if you need further help or documentation. I would recommend using a different time constant for each mean wind speed.

We agree that the dynamic inflow model is an important correction model and that a fair comparison of the aerodynamic models should include it (it was also suggested by referee 1). With your help, we were able to re-run the BEM simulation with the DBEMT model. Thank you again for your support. We have updated the results and analysis in the new version of the manuscript.

- Aerodynamic differences. As you mention, the elastic and servo parts are the same for both models, the only thing affecting the results are differences of the aerodynamic model. The paper follows a nice scientific approach, yet it seems that there is a gap between the stiff and steady simulations, and the elastic unsteady simulations. It would be valuable to investigate the key aerodynamic differences between the models, using stiff unsteady simulations (e.g. performing a sweep at different yaw angles, studying response to wind steps, or looking at azimuthal variations of inductions similar to fig 13). If possible, these results could be compared to CFD, measurements, or, other BEM implementations. Such results could then be used to interpret the results of the full aero-servo-elastic simulations. This again would require more work, but I think it would be valuable to focus on this first, and maybe present the fatigue analyses in a separate paper... Even if you chose to stay with the current structure, I think it is important if it is stressed that the differences are between two specific implementations, the BEM from OpenFAST and the free vortex wake from QBlade. In light of this, I would think you might want to revise the title to highlight that the results are specific to these two implementations, unfortunately, the title would be less catchy.

The goal of this paper is to analyze the effect of the different aerodynamic models on turbine behavior and loads using industry standard methods so that the results can be easily interpreted by the industry. This, in our belief, is also the main and novel contribution of the paper: quantifying the loads using industry metrics when these two aerodynamic methods are used. A significant amount of previous published works examines the differences of different BEM implementations and LLFVW codes for several, more idealized, cases. See for example the references Marten et al. 2015, Saverin et al. 2016a, Hauptmann et al. 2014 and Boorsma et al. 2016. In fact, the results from many of the cited papers were used in the present manuscript as a basis to analyze and understand the source of the differences in the loading. To emphasize that the conclusions are only valid for the used codes, we changed the manuscript title and included the information in the introduction, discussion and conclusions (referee 1 also suggested this).

- Length of the paper: Despite the careful and valuable analyses and discussions, I believe the paper could be considerably reduced in length. Here are some ideas to reduce the length: The literature review and presentation of the models can be significantly reduced. The results of section 5 are a bit repetitive. You may consider presenting one of the plot for one sensor, and then simply focus on the key conclusions that you drew for the other sensors (maybe summarizing them in a table). You are thorough

in your analyses, and I believe the reader will trust your conclusions without having to see the plots. Also, I would think you can remove some text that describe the figures, and move more rapidly to their discussions. The reader might get lost in such a level of details, and I would advise to focus more on the story and key conclusions of the paper.

Based on your recommendations, we reduced the length of the literature review and the model description of the BEM method. We also reduced the discussion section to focus on the story and removed the PSD plots for the yaw bearing sensors. We still included the discussion for the tower base and yaw bearing sensors. They are often left out in the referenced literature and we saw that the differences in aerodynamic models affected them in other ways than the out-of-plane blade root bending moment. Including them in the discussion gives the reader a more complete picture of the effects of the aerodynamic models.

I enclose some specific comments below. I hope that addressing these general and specific comments will improve the quality of the paper. Clearly a lot of work has been done, and I would be happy to review a revised version of the manuscript. Good luck for the work

p1 l1: "state of the art" might not be appropriate -> "common" maybe (CFD would be state of the art)

Changed the sentence so that 'state of the art' does not appear

p1 l3: BEM does not only simplify the rotor-aero (also wake and inflow). To some extent the rotor aero are the same for BEM and FVW

Removed the sentence on rotor aerodynamics to avoid misinterpretations.

p1 l19: the "wake memory" effect need to be included for this study, otherwise the results won't be fair

We updated the results so that the BEM calculations now include the wake memory effect. Sections 4, 5, 6 and 7 have been modified accordingly.

p3: very nice literature review, but quite long, maybe a page could be removed.

The literature review has been shortened.

p5 l13: "yawed-inflow condition" -> or "yawed and tilted conditions"

Removed this part of the model description because of the suggestion of reducing the length of the paper

p5 l24: the dynamic wake model is the model of {\O}ye

Included the name Oye in the description.

p5: the model description could be condensed to a smaller list with references to shorten the paper.

Shortened the description of the engineering models and referenced to the AeroDyn theory manual

p6: Can you detail the core model you are using? In particular, please mention how you determine the core size of the bound and wake vorticity? How are these parameters determined as function of the discretization?

Added the information of the vortex core size the model in Section 2.1.2.

p8 l3: "in the rotor plane": It can be argued that the main issue comes from the fact that the annuli are assumed to be independent, not so much that the momentum balance was determined in a plane.

Changed the sentence so that the emphasis is on the annuli and not on the plane.

p8 l9: ElastoDyn does not rely solely on Euler-Bernoulli beam theory, it also includes corrections to account for geometric non-linearities.

Added the sentence: 'It also includes corrections to account for geometrical non-linearities.'

p10 l1: Feel free to contact me if you want more information on the DBEMT model. I believe this model needs to be used for the current study

Thank you for the offer and the assistance you provided, with your help we now included this effect in the study.

p10 l14: It would be valuable to present aerodynamic performances of the FVW&BEM at exactly the same operating conditions (RPM/Pitch) instead of using the controller. Presenting radial distribution of axial and tangential inductions along the span at different operating point will reveal the aerodynamic differences between the two models. The inductions are the main variables of lifting-line codes (such as BEM and LLFVW).

We agree that such a study would give valuable information. In the present study, we are more interested in the aero-servo-elastic load calculations than in the purely aerodynamic analysis. Figures 3 and 4 show a comparison of the integral values by comparing thrust and power calculated with both codes. On this basis, we expect the loads on the considered sensors to be comparable. In fact, a comparison of the loading along the blade using both codes for different steady state wind situations can be seen in the references Saverin et al. 2016b and Perez-Becker 2018. The power coefficient calculations in Figure 3 also compare to the ones done with HAWCStab2 and EllipSys3D. These calculations were done with prescribed pitch angles and rotor speed. Our results agree with the ones obtained from other codes quite well. This shows that including the interaction with the controller does not distort significantly the aerodynamic results compared to prescribed conditions. The steady state values of thrust and power between QBlade and OpenFAST in Figure 4 also agree fairly well so that there will be no significant offset when determining the extreme and fatigue loads of the considered sensors. As for the operating conditions, the pitch angles of the blade are almost identical for both codes and the maximum difference of the rotor speed is 5.5%. So again, we do not expect that the results from simulations with prescribed pitch and rotor speed would give qualitatively different results.

In addition, a main result of the steady wind simulations is that the inclusion of the wake coarsening methods had a small influence on the aero-servo-elastic response of the turbine in steady wind conditions. If we fixed the rotor speed and pitch angle, we would be forcing the coincidence of the turbine states and the effect of the wake coarsening methods on the aero-servo-elastic behavior of the turbine would be more difficult to assess. This would also make it more difficult to communicate the

aforementioned result. Including an additional analysis of the induction factors will extend an already long paper without contributing to the main findings, which are unsteady by nature.

p11 tab2: You may consider using "steady" instead of "constant"

Changed the description to 'steady wind'

p11 l20: "Purely aerodynamic calculations" -> you could replace by "CFD" or something similar maybe?

The purely aerodynamic calculations refer to the simulations done with the parameters under the column 'aerodynamic calculations' of Table 2. We did not use CFD in this paper. We modified the paragraph so that it states more clearly that we are comparing simulations that used the parameters in Table 2.

p13 tab3: The fact that only DLC1.2 is used might indeed make the comparison of the ultimate load difficult, you can consider using a gust case. This would avoid the some of the discussions of section 6 related to the different responses of the two models, where maxima occur at different simulations.

The extreme load evaluation done on the power production simulations corresponds to the DLC1.1 from the IEC 61400-1 ed3 standard. Since the goal of this paper is to show the differences of BEM and LLFVW aerodynamic models using methods following those used in the industry, we decided to do the extreme load evaluation with this reduced load set according to the standard. We agree and mention in our conclusions that DLC1.1 is not a load case group that usually includes the design driving extreme loads of the turbine. Yet including and analyzing the results gives an insight on the mechanisms that cause the differences in the extreme loads. Using a gust case to analyze the extreme loading is an arbitrary choice that most likely does not reflect the situations that contributed to the design driving extreme loads of the turbine. These are the DLC1.3 and 6.2 (see reference Bak 2013) and will be analyzed in future work. Please see the reference Hauptmann 2014 for a comparison of a BEM and a LLFVW code in a gust case.

p13 l16: Though I like the extra information on the plot, you can maybe mention that 6 points are likely not statistically significant.

The statistical information of each sensor was taken from all the data points of the six time series for a given wind speed bin. That is 90000 data points per wind speed bin per sensor. We included some more explanation in this sentence.

p17 l5: How do your results compare to other studies?

Included the numerical values of the references. We also included a statement that a direct comparison is not possible due to the small amount of cases and simulation time used in the referenced studies.

p19 l17: The main effect there is the small amplitudes of the inductions, not the pitch angle. Probably what is meant here is that the angle of attack is mainly composed by the rotational speed and the undisturbed wind, since the inductions are small, and hence the rotor performances are not strongly affected by the aerodynamic model.

The idea of this sentence is to explain the global effect of the wake aerodynamics in a more physical way than with the induction factors. Since the induction factors are calculated in an iterative way, they are affected by the pitch angle of the blade. The latter changes the local inflow angle of the blade element, hence affecting the axial induction. To avoid misunderstandings and to accommodate the remarks of the referee 1, we changed the paragraph to state the message more clearly.

p20 l7: Both the BEM implementation from OpenFAST and vortex code have limitations in yawed inflow. For the vortex code, the "wake is going up" in skewed inflow (I discuss this in a paper entitled "Aeroelastic large eddy simulations using vortex methods: unfrozen turbulent and sheared inflow"). On the other hand the limitations of the OpenFAST BEM code are more inherent the implementation choices (coordinate systems used for the axial and tangential inductions, and choice of determination of the wake skew angle), these choices are made differently in different codes. You may want to mention this somewhere in the paper. This is why in my general comments I mention that the observations made are specific to the implementations, and it would be highly valuable to present some "stiff" simulations comparing some key aerodynamic components of the models (against, CFD, measurements, or other BEM codes).

We mention in the new manuscript title ("Is the Blade Element Momentum Theory overestimating Wind Turbine Loads? - An Aeroelastic Comparison between OpenFAST's AeroDyn and QBlade's Lifting Line Free Vortex Wake Method"), introduction, discussion and conclusion that the comparison is only of two particular aerodynamic codes. We also mention the limitations of the LLFVW in sheared inflow and the different existing implementations of BEM codes in the discussion section 5.3.

p22 l1-4: How did you determine the time constant for DBEMT here? This might need to be adapted since it does not filter the high frequencies enough.

The time constant for DBEMT was determined automatically by OpenFAST with an appropriate parameter in the AeroDyn input file.

p23 l12: Is wake memory actually included in Fig 10d? I might have missed it. Or is the figure reference wrong maybe?

Figure taken out because OpenFAST is now run with only the DBEMT option

p27 l2: "have a mean value of 0" -> "have been adjusted to a mean value of 0", maybe?

Figure taken out because OpenFAST is now run with only the DBEMT option

p27 l15: DBEMT will also introduce a phase shift, maybe similar to the LLFVW code, or not.. p27 l24:

Figure taken out because OpenFAST is now run with only the DBEMT option

P27 l24: "In last section" -> "In the previous section", or, "In section ...

Changed beginning of sentence

p30 l21-29: The comparison of the two simulation might be too difficult (or "anecdotic") since the wind turbine is indeed a highly non-linear system, and both aerodynamic models are behaving quite differently here. I would recommend using a more deterministic case like a gust for the study. Figures 14-15 are still interesting and valuable.

Aeroelastic turbulent wind load calculations are indeed more difficult to analyze than deterministic load cases. Yet we believe that analyzing the former cases gives a better understanding of the influence of the aerodynamic models on realistic aero-servo-elastic scenarios. By analyzing the interaction of the aerodynamic code, the structural code and the turbine controller in an 'anecdotic' way, we give a plausible explanation on the observed differences in turbine behavior, We agree that this analysis is in no way complete but it does give an insight into the driving phenomena that cause the difference. This helps us understand the mechanisms in which the different aerodynamic codes affect the aero-servo-elastic mechanisms in those particular load cases.

We updated the explanations and figures is Section 6.3 so that the results from BEM calculations now include the DBEMT option.

---

## Referee Report (RR1)

Thank you for addressing my comments and performing the extra work. The differences in loads are quite significant, a message that is certainly worth communicating!

---

## Author Response (AR2)

**Editor (K. Dykes)**

Good job addressing reviewers' comments and nice overall work. Minor edits suggested below. Generally, the introduction can use some additional attention, but the paper is otherwise in good shape.

Thank you for your review and appreciation of the paper. Please find our responses and changes to the manuscript below your comments in blue.

Note, author name order reversed at present. Hopefully formatting from journal will correct.

⇨ We changed the order of the author names.

Section 1

Line 9 – what do you mean scale accordingly? Also, it is not good to cite an entire textbook – this is lazy. Should use a specific reference to content within the text or cite a particular paper on scaling of blade loads with length (there are several nice journal articles on blade scaling of loads and mass with length).

By this sentence we meant that the loads of individual components of the wind turbine scale as a function of the rotor diameter. This function depends on the actual component being considered. For this reason we mentioned 'accordingly'.

⇨ Changed the sentence so that it is clear that the loads scale as a power of the rotor diameter. We also added the pages of cited book.

Line 11 – you mean over conservatism correct? Differences is not specific – differences from what?

Yes that is what we meant.

⇨ Changed sentence to explicitly mention the over-conservatism on the load estimation.

Line 13 – again, citing textbooks without specific reference to in text content is not adequate for a journal quality paper. Go to the source and cite key peer reviewed literature. For instance, as you are using FAST, you might cite the papers that support the formulation currently in AeroDyn.

⇨ We added the pages in the textbooks where the blade element momentum theory is explained

Line 15 – be more specific. In the abstract line 4, you saw that BEM will over estimate loads, this is only self evident if the engineering corrections to the model are intentionaly conservative. How and why is BEM predicting wrong? Why aren't the engineering corrections good enough?

⇨ We extended the paragraph to explain more in detail why BEM can predict inaccurate loads. Basically, the inaccuracies arise when the turbine operates outside the conditions in which the correction models were tuned and tested. We added some examples of these conditions including citations where these issues are a main source of load differences.

Line 22 – again, not self evident that using a more accurate method leads to lower deign loads… this is only the case if conservatism to counteract the uncertainty of the lower fidelity model. Expand on this point within the preceding paragraph.

⇨ Expanded the whole paragraph to explain the point further. The paragraph now briefly explains why BEM methods can have inaccuracies in load predictions (your previous point). We also explain that the design loads can be reduced by either better load predictions with the LLFVW method or indirectly by reduced safety factors applied to the design loads.

Section 2.2

Line 20 – okay explanation but not great. Might speak to the limitations of this decision a bit more – for instance, what if you used a code like beamdyn with the respect bem and llfvw codes? How would that influence the comparison? You cover it a bit but not fully in the preceding paragraph

⇨ We expanded the section by addressing other ways in which structural models with higher accuracy affect the loads and very likely magnify the load differences coming from different aerodynamic methods. In order to quantify this, a separate study would be needed. We point this out in the conclusions. We also included a reference to a study that compares several combinations of ElastoDyn, BeamDyn, AeroDyn and QBlade'ss LLFVW method. It shows that changing the structural model has large influence on the loads.

Line 13 – again okay explanation, but might speak to the limitations of this decision

⇨ Expanded on the limitations of our decision: we expect the controller to work suboptimal in the LLFVW simulations, because it was optimized based on BEM calculations.

Section 3

Line 11-13 – why is qblade predicting higher rotor speed? Lacking full explanation

Basically, the full explanation is that the induction factors and the loads are being calculated with fundamentally different aerodynamic models. It should be no surprise that there are small differences in the predictions of the steady state variables such as the rotor power and the rotor thrust. The higher rotor speed comes from the higher power coefficient calculated with QBlade. Section 3 expands on this by including several comparisons with other codes. Higher power coefficients for the same wind speeds are also seen when the CFD-based code EllipSys3D is compared to the BEM-based code HAWCStab2. Differences of 4-5% in rotor power between different BEM-based and vortex-based codes are also reported in the reference (Madsen et al. 2012), which we included and compared to in this section.

⇨ Added a brief explanation about the higher rotor speed in this section.

Section 4

Line 2-3- Can you justify the selection of these DLCs? Here and upfront in the introduction?

The DLC 1.2 accounts for most of the fatigue damage the turbine components see. Including this DLC group limits the number of calculated load cases while still giving a good estimate of the lifetime fatigue loads of the turbine components.

⇨ Added the explanation of our selection in this section and briefly in the introduction.

Section 4.1

Line 11-12 – inadequate explanation of sensor selection

⇨ Added detail to the sensor selection section to explain why we chose them.

Figure 5 – figures are quite small and it is hard to see the differences. Consider enlarging them a bit

We agree that this figure is somewhat small in the paper. The actual figure is much larger and the differences are easier to see. Yet from our understanding of the template, there are only two possible figure sizes. This is already the larger figure size (two column). Perhaps the size can be increased in the final formatting from the journal. Including the figures as individual figures would take too much space and give these figures too much apparent importance. In our opinion, these figures should only give an overview of the turbulent load calculations using integrated rotor quantities. One of the results is that for these integrated quantities the differences between the models are small. The source of the load differences is therefore more subtle.

⇨ Added a sentence to explicitly say that the differences in these integrated quantities is small.

Section 4.2

Overall lacking critical analysis of the differences of the results. These performance differences will obviously lead to differences in loads. Why is it happening? Need to tie this back to the physics or the model differences or nuances of the controller… hint at what is coming in the following sections.

This section is supposed to present an overview of the results. Including already an analysis of why this is happening in this section is not straight forward and would need much explanation that is included later in the paper. We believe that the current structure of the paper in which the differences are presented, quantified and latter discussed and explained is relatively intuitive and helps improve the readability of the paper. Including extensive analyses at the beginning while the data is being presented would not improve the readability of the paper.

⇨ Expanded the section to include some preliminary analysis of the results and a better transition to the next sections.

Section 7

Good acknowledgement of future work – can you speak a bit more to the broader impacts of this effort? Its value?

⇨ We added a sentence tying the conclusion to the initial motivation of this study: By obtaining lower fatigue loads by adopting a higher order aerodynamic model, we see real potential of this aerodynamic method to reduce the design loads of future wind turbines. But more work needs to be done before we can say this as a general conclusion.